# Molecular Basis of BRAF Inhibitor Resistance in Melanoma: A Systematic Review

**DOI:** 10.3390/ph18081235

**Published:** 2025-08-21

**Authors:** Ilaria Cosci, Valentina Salizzato, Paolo Del Fiore, Jacopo Pigozzo, Valentina Guarneri, Simone Mocellin, Alberto Ferlin, Sara Mathlouthi, Luisa Piccin, Mariangela Garofalo

**Affiliations:** 1Unit of Andrology and Reproductive Medicine, Department of Medicine, University of Padova, 35122 Padova, Italy; 2Medical Oncology 2, Veneto Institute of Oncology IOV-IRCCS, 35128 Padova, Italy; valentina.salizzato@iov.veneto.it (V.S.);; 3Soft-Tissue, Peritoneum and Melanoma Surgical Oncology Unit, Veneto Institute of Oncology IOV-IRCCS, 35128 Padova, Italy; 4Department of Surgical, Oncological and Gastroenterological Sciences (DISCOG), University of Padova, 35138 Padova, Italy; 5Department of Pharmaceutical and Pharmacological Sciences, University of Padova, 35131 Padova, Italy; sara.mathlouthi@unipd.it (S.M.); mariangela.garofalo@unipd.it (M.G.)

**Keywords:** melanoma, BRAF mutation, V600E, MAPK pathway, targeted therapy, BRAF inhibitors (BRAFi), MEK inhibitors (MEKi), drug resistance

## Abstract

**Background**: Melanoma, the deadliest human skin cancer, frequently harbors activating BRAF mutations, with V600E being the most prevalent. These alterations drive constitutive activation of the MAPK pathway, promoting uncontrolled cell proliferation, survival, and dissemination. The advent of BRAFi and MEKi has significantly improved outcomes in BRAF V600-mutant melanoma. However, therapeutic resistance remains a major clinical barrier. **Methods**: This review integrates recent findings from preclinical and clinical studies to delineate resistance mechanisms to BRAF-targeted therapy. It categorizes resistance into *primary* (intrinsic), *adaptive*, and *acquired forms*, and analyzes their molecular underpinnings, including genetic and epigenetic alterations, pathway reactivation, and microenvironmental interactions. **Results**: Primary resistance is linked to pre-existing genetic and epigenetic changes that activate alternative signaling pathways, such as PI3K-AKT. Adaptive and acquired resistance includes secondary BRAF mutations, pathway redundancy, phenotype switching, and immune and stromal interactions. High-throughput sequencing has revealed novel mutations, including NRAS, NF1, and PTEN alterations, that contribute to resistance. **Discussion**: Understanding the multifaceted nature of resistance is critical to improving outcomes in advanced melanoma. This review highlights emerging strategies to overcome resistance, including combinatorial therapies, metabolic targeting, and biomarker-driven approaches, aiming to inform future therapeutic development and precision oncology strategies.

## 1. Introduction

### 1.1. Rationale

Melanoma is an aggressive malignancy originating from melanocytes, the pigment-producing cells responsible for protecting the skin from ultraviolet radiation [1]. Although mortality rates are decreasing in some regions, the global incidence of melanoma continues to rise, particularly among younger populations [2]. The prognosis remains favorable in early stages but becomes significantly worse in patients with loco-regional or distant metastatic disease, with 5-year survival rates dropping to 75% and 35%, respectively [3].

A major therapeutic breakthrough has been the identification of activating BRAF mutations, most notably the V600E variant. These mutations are responsible for the continuous activation of the downstream MAPK signaling, resulting in tumor development through cell cycle enhancement and inhibition of apoptosis [4,5,6,7].

Thanks to the discovery of BRAF V600 mutations, molecularly targeted inhibitors were developed. Significant improvements in progression-free survival (PFS) and overall survival (OS) have been observed compared to the previous standard of care, chemotherapy [8,9,10,11,12].

The clinical introduction of BRAF inhibitors (BRAFi), and later their combination with MEK inhibitors (MEKi), has substantially improved outcomes in patients with BRAF-mutant melanoma, outperforming previous chemotherapy-based treatments.

However, despite initial responses, resistance to BRAFi therapy remains a formidable clinical challenge. A proportion of patients exhibit primary resistance, failing to respond from the outset, while others develop acquired resistance after a period of clinical benefit. Mechanisms of resistance are complex and multifactorial, involving genetic and epigenetic alterations, pathway reactivation (such as MAPK and PI3K/AKT), metabolic rewiring, tumor microenvironment interactions, and immune escape. Although individual mechanisms have been extensively studied, a comprehensive and integrative understanding is still lacking [13].

### 1.2. Objectives

This review aims to systematically explore the molecular basis of resistance to BRAF-targeted therapies in melanoma. We categorize resistance mechanisms into *primary*, *adaptive*, and *acquired*, highlighting the interplay of signaling pathways, metabolic networks, and immune dynamics. In addition, we discuss current and emerging therapeutic strategies, including rational combination therapies and biomarker-guided approaches, to overcome resistance and improve clinical outcomes in advanced melanoma.

## 2. Methods

This is a systematic review of studies reporting mechanisms of resistance to BRAF-targeted therapy in melanoma patients treated with BRAFi. The review was conducted in accordance with the Preferred Reporting Items for Systematic Reviews and Meta-Analyses (PRISMA) 2020 guidelines. A comprehensive investigation of the literature was performed across major databases, and relevant studies were screened and selected based on predefined inclusion criteria. Data extraction and synthesis focused on categorizing resistance as *primary* (intrinsic), *adaptive*, or *acquired*.

### 2.1. Search Strategy

MEDLINE/PubMed, EMBASE, and SCOPUS were systematically exploited to identify eligible studies. The search was performed with language restrictions (including only studies in English) up to June 2024. In PubMed, EMBASE, and SCOPUS the following search strategy was used: “BRAF resistance” AND “mechanism” AND “melanoma”, and it was tailored to fit other electronic sources. The lists from each source were combined (*n* = 2192). Two investigators (IC, PDF) separately reviewed search results, screened titles/abstracts, excluded duplicate articles (*n* = 1935), and removed those outside the scope of the review (*n* = 140). The full-text versions of all potentially eligible articles were examined to remove those not fulfilling the inclusion criteria. Finally, the reference lists of the included articles were manually explored to identify any further studies of interest. Disagreements were solved by consensus with a third investigator (MG). Cohort studies, case-control studies, and review reporting information on BRAF resistance in melanoma treatment from 2012 to 2024 were eligible for inclusion. Studies not including human subjects were excluded.

### 2.2. Data Collection

Two investigators (IC, PDF) independently extracted relevant topics from the selected articles. For each article, they recorded the type of resistance, immune-related mechanisms, and alterations in metabolic pathways. A third investigator (MG) verified the extracted data for accuracy and consistency.

## 3. Results

### 3.1. Search Results

The search identified 257 non-duplicated articles. After careful analysis, 140 articles were excluded based on title/abstract. Of these, 131 were excluded due to a different design (*n* = 50), a different topic (*n* = 77), or different participants (*n* = 4). Nine were excluded because the full-text version could not be found. Finally, 117 articles were included in the synthesis (Figure 1).

### 3.2. Study Characteristics

The key landmark studies on BRAF resistance are categorized by type of resistance: *primary resistance*—including genetic, epigenetic, and transcriptomic mechanisms; *secondary resistance*—both adaptive and acquired; as well as *immune-related mechanisms* and alterations in metabolic pathways.

## 4. BRAF Overview

In 2002, Davies et al. demonstrated that BRAF-mutated proteins were present in 66% of malignant melanomas and, with low frequency, in other human cancers [8]. In 2015, the Cancer Genome Atlas Network (TCGA) classified cutaneous melanoma into four primary molecular subtypes: BRAF-mutant (52%), RAS-mutant (28%), NF1-mutant (14%), and Triple-WT. The three mutant subtypes result in the constitutive activation of the MAPK pathway, which is implicated in tumor cell proliferation and survival but also in invasion and metastasis, that regulates key cellular functions such as growth, proliferation, differentiation, and apoptosis through a sequence of intracellular proteins [9]. The BRAF gene, located on chromosome 7 (7q34), encodes a 94 kDa serine/threonine protein kinase composed of 766 amino acids central in the MAPK pathway.

This cascade regulates key cellular functions such as proliferation, differentiation, and apoptosis through a sequence of intracellular proteins. BRAF, along with ARAF and CRAF, acts as a MAPK kinase kinase (MAPKKK) and is usually activated by GTPase proteins, like RAS, following stimulation by cell surface receptors such as the Epidermal Growth Factor Receptor (EGFR) or KIT. Various other stimuli can also activate BRAF. In physiological conditions the pathway is initiated when growth factors or cytokines bind to their respective receptors on the cell surface. This event activates the small GTPase protein RAS, which in turn recruits RAF proteins kinases ARAF, BRAF, and RAF1 to the cell membrane. Upon activation, BRAF phosphorylates and activates MEK1/2 kinases, which then phosphorylate and activate ERK1/2 kinases. The activated ERK1/2 translocate to the nucleus and regulate the expression of genes involved in cell proliferation, differentiation, and survival [6].

The BRAF protein has three conserved regions: CR1, which contains a RAS-binding domain (RBD) and a cysteine-rich domain; CR2, a serine/threonine-rich domain with a 14–3–3 binding site; and CR3, the catalytic domain of the serine/threonine protein kinase. For BRAF to function properly, it must dimerize and form complexes with MEK, facilitated by 14–3–3 proteins involved in both active and inactive states of BRAF signaling. Once activated, BRAF phosphorylates MEK, which in turn phosphorylates ERK1 and ERK2. These ERK proteins, as the final effectors of the pathway, dimerize and enter the cell nucleus to activate transcription factors like c-Jun and c-Myc. The primary functions of this signaling pathway are to control cell progression and to regulate apoptosis [10].

Approximately 50% of melanomas exhibit BRAF V600-mutations, involving the substitution of valine in position 600 with glutamic acid (V600E, up to 80–90% cases), lysine (V600K, 10–21%), arginine (V600R, 5%) or aspartate (V600D < 5%) [11]. Within the most frequent mutation, the replacement of the non-polar valine (V) with a negatively charged glutamic acid (E) at position 600 (V600E) locks the protein in a phosphomimetic activated state with a 480-fold enhancement in kinase activity compared to the native protein’s catalytic conformation. As a consequence, persistent activation of the downstream MAPK signaling and suppression of negative feedback mechanisms occur, granting RAS independence, transformative ability, and a 10-fold increase in cell growth [8]. Moreover, BRAF V600E seems to be also implicated in tumor dissemination and invasion by (*i*) activating the GTPase Rho protein that causes reorganization of the cytoskeleton; (*ii*) downregulating phosphodiesterase 5A (PDE5A) with increase of cGMP levels; (*iii*) enhancing cortactin and Exo70 phosphorylation mediated by ERK, with consequent actin cytoskeleton reorganization and metalloproteinases exocytosis (Figure 2). Thus, the identification of BRAF as a potential therapeutic target has revolutionized and broadened the spectrum of therapeutic options for melanoma management, with a dramatic impact in survival, while paving the way for contemporary targeted therapy approaches as well for other human malignancies in the era of precision medicine [12].

## 5. Targeted Therapies: Drug Mechanisms and Treatment Options

BRAF inhibitors (vemurafenib, dabrafenib, and encorafenib) work by binding to the ATP-binding site of the mutated BRAF V600 kinase. This site is crucial for the enzyme’s activity, as it allows the kinase to obtain the phosphate groups needed to phosphorylate and activate MEK, the downstream protein in the MAPK signaling cascade. By occupying this site, BRAFi prevent ATP from binding, thereby blocking BRAF’s ability to phosphorylate MEK. As a result, ERK, the final kinase in the pathway, is not activated and cannot translocate to the nucleus to promote gene expression, cell proliferation, and survival. Inhibiting ERK leads to reduced tumor cell proliferation and increased apoptosis, or programmed cell death. MEK inhibitors (cobimetinib, trametinib, and binimetinib) directly inhibit MEK1 and MEK2 proteins by preventing the phosphorylation of ERK, another key protein in the MAPK pathway [13,14].

Current therapies for BRAF-mutated melanoma include three BRAFi/MEKi combinations: vemurafenib + cobimetinib, dabrafenib + trametinib, and encorafenib + binimetinib (Table 1). The activity and efficacy of these three combinations are generally considered similar, despite the absence of direct comparative studies. However, encorafenib has shown more prolonged and potent pharmacodynamic activity compared to vemurafenib and dabrafenib, with potentially favorable clinical implications [15].

In 2011, the introduction of BRAFi monotherapy in melanoma patients led to rapid and consistent responses, with an objective response rate (ORR) of approximately 50%, compared to chemotherapy, which had been the standard of care and showed an ORR of just 5–6%, a median overall survival (OS) of 7.5 months, and a 5-year survival rate of 6% [16].

Nevertheless, despite initial effectiveness, responses were short-lived, mainly due to the paradoxical reactivation of MEK, the downstream protein in the MAPK cascade [17].

To overcome this limitation, combination strategies with BRAFi and MEKi were developed. Compared to monotherapy, these combinations showed increased efficacy, with an ORR of 63–70% versus 50–53%. Additionally, the combination therapy led to a significant improvement in median progression-free survival (mPFS) from 7.2–9.6 months to 11–14.9 months and median overall survival (mOS) from 17.4–23.5 months to 22.5–33.6 months [18,19,20,21]. Moreover, combination therapies were associated with a lower risk of secondary neoplasms, particularly primary skin tumors, compared to BRAFi monotherapy (4–11% vs. 13–24%) [20] A detailed analysis of the safety profiles of these combinations, although limited by the lack of direct comparison studies, revealed no significant differences in the overall risk of adverse events (AEs) of any grade or serious AEs. However, grade 3 AEs (based on CTCAE criteria) were reported more frequently in patients treated with vemurafenib/cobimetinib (71%) compared to other combinations (52–58%). Beyond common side effects—such as fatigue, nausea, and ocular toxicity associated with MEKi—each combination has a distinct toxicity profile. For example, pyrexia is more frequently observed with dabrafenib/trametinib and is the most common cause of dose interruption or discontinuation. Photosensitivity is more common in patients receiving vemurafenib/cobimetinib, whereas venous thromboembolism occurs more frequently in those treated with encorafenib/binimetinib [22].

Although adverse effects are frequent, they are generally rapidly reversible and do not cause long-term sequelae. However, the most critical limitation of these therapies—despite their ability to induce fast and significant responses—remains treatment resistance. Approximately 20% of patients with BRAF V600-mutant melanoma do not respond to targeted therapy (primary resistance), and secondary resistance develops in a significant proportion of cases. After five years of follow-up, over 70% of patients are estimated to experience disease progression despite targeted treatment [23].

## 6. Type of Resistance

Despite a prompt tumor response and disease control achieved by BRAFi plus MEKi, disease progression can emerge from starting therapy within 6–8 or 11–14.9 months when BRAFi is used as a single agent or in combination with MEKi, respectively, due to the development of resistance mechanisms [24].

The mechanisms of resistance are divided into three main categories: *primary* or intrinsic resistance, where the tumor does not respond to therapy from the beginning due to genetic mutations; *adaptive* resistance, which involves an initial response followed by drug tolerance not due to mutations, often reversible; and *acquired* resistance, which emerges in an advanced stage and is characterized by drug tolerance caused by mutations, which is usually irreversible [25,26,27].

### 6.1. Primary (Intrinsic) Resistance: Genetic Mechanisms of Resistance

Primary resistance is characterized by several genetic mutations responsible for a lack of response to BRAFi treatments. This type of resistance involves several key mechanisms (Table 2), including the loss of the phosphatase and TENsin homolog (*PTEN*) gene, amplification of the cyclin D1 (*CCND1*) gene, Cancer Osaka Thyroid kinase (COT, MAP3K8) overexpression, loss of the neurofibromin 1 (*NF1*) tumor suppressor gene, mutations in Ras-related C3 botulinum toxin substrate 1 (RAC1P29S), and the disruption of the ubiquitin-specific peptidase 28 (USP28-FBW7) complex.

#### 6.1.1. Loss of the Phosphatase and TENsin Homolog (*PTEN*) Gene

The phosphatase and TENsin homolog (*PTEN*) gene, firstly identified in 1997 [28], is a tumor suppressor gene that negatively regulates the phosphoinositide 3-kinase-protein kinase B (PI3K-AKT) signaling pathway, one of the main pathways controlling cell growth, survival, and proliferation.

PTEN acts as a lipid phosphatase, de-phosphorylating phosphatidylinositol 3,4,5-trisphosphate (PIP3) to phosphatidylinositol 4,5-bisphosphate (PIP2). This process reduces PIP3 levels in the plasma membrane, limiting the activation of protein kinase B (AKT) and downstream signaling pathways that promote cell growth and survival. Furthermore, PTEN also has a role in regulating focal adhesion kinase (FAK), a tyrosine kinase involved in the formation of focal adhesions and cellular structures that mediate cell anchorage to the extracellular matrix, and it is crucial for cell migration and invasion. Indeed, through FAK dephosphorylation, PTEN inhibits the cells’ ability to form stable focal adhesions and migrate, thereby limiting their invasiveness and capacity to spread, which is essential for preventing tumor metastasis. Additionally, PTEN dephosphorylates Shc (Src homology 2 domain-containing transforming protein), preventing signal transduction through the MAPK pathway. This leads to the suppression of MAPK signaling stimulated by growth factors, thereby limiting tumor proliferation [29].

Stahl et al. highlighted PTEN loss as a critical factor in melanoma development by promoting cell survival and reducing apoptosis. To investigate the role of PTEN loss in melanoma, researchers introduced an intact chromosome 10 into melanoma cells lacking PTEN, restoring its expression to normal levels. This reintroduction slowed tumor growth in mice, unless PTEN was later deleted or inactivated. PTEN loss was found to activate AKT, suppressing apoptosis and promoting cell survival, which favors tumor formation. Conversely, PTEN expression inhibited AKT, increasing sensitivity to apoptosis. The study confirms that PTEN loss is crucial for melanoma development by reducing apoptosis and supporting tumor growth [30]. In primary resistance, cells lacking PTEN continue to activate the PI3K/AKT pathway, thus bypassing the inhibition of the MAPK/ERK pathway. This phenomenon has been observed in several melanoma cell lines with the BRAF V600E mutation, where loss of PTEN significantly reduces the efficacy of PLX4720.

Another mechanism by which loss of PTEN contributes to resistance is by suppression of BIM protein expression.

BIM (BCL-2-interacting mediator of cell death) is a pro-apoptotic protein of the BCL-2 family that plays a key role in inducing apoptosis in response to stress or cell damage signals. When the PI3K/AKT pathway is hyper-activated, as happens in the absence of PTEN, a reduction in BIM expression is observed, decreasing the ability of cells to undergo apoptosis [31].

#### 6.1.2. Role of CCND1 Amplification

The *CCND1* gene encodes cyclin D1, a crucial regulator of the cell cycle. The cyclin D protein participates in various intracellular pathways, both CDK4/6-dependent and independent, including the promotion of cell proliferation, regulation of mitochondrial functions, DNA damage repair and cell migration. Throughout the cell division cycle, cyclin D levels and its subcellular localization fluctuate in an oscillatory manner, responding to proliferative and mitogenic signals. When growth factors bind to their respective tyrosine kinase receptors (such as insulin-like growth factor receptor (IGFR), c-kit, or fibroblast growth factor receptor (FGFR)), the RAS/RAF/MEK/ERK signaling pathway is activated, triggering the transcription of CCND1 and subsequently increasing cyclin D expression in the early G1 phase.

Cyclin D then forms a heterodimer with CDK4/6, and this active complex translocates into the nucleus. There, in cooperation with the cyclin E/CDK2 complex, it induces hyperphosphorylation of three growth-suppressive proteins: the tumor suppressor retinoblastoma protein (RB), retinoblastoma-like protein 1 (p107), and retinoblastoma-like protein 2 (p130). This hyperphosphorylation leads to dissociating the E2F transcription factor from RB, allowing E2F to activate genes necessary for DNA replication, thereby facilitating the cell cycle’s transition to the S phase. It has been reported that in a subset of melanomas (<8%), the *CCND1* gene is amplified, leading to an overproduction of cyclin D1, which enables an aberrant proliferation, diminishing the effectiveness of treatment, and generating a therapy resistance mechanism [32].

#### 6.1.3. COT (MAP3K8) Overexpression

COT, also known as mitogen-activated protein kinase kinase (MAP3K8), is a serine/threonine kinase that plays a critical role in regulating the MAPK/ERK pathway. COT can activate this pathway by directly phosphorylating MEK1 and MEK2, the kinases that subsequently activate ERK1/2, independently of upstream signaling molecules such as RAF, including BRAF. This allows the reactivation of ERK, leading to the restoration of cell signaling that promotes proliferation and survival. Upregulation or activation of COT allows tumor cells to bypass blocked BRAF and to fuel MAPK/ERK pathway cascade, reactivating cell proliferation. This type of mutation is observed in about 1.5% of melanoma patients and is particularly common in Spitz nevi, occurring in about 33% of cases [33].

#### 6.1.4. Role of Loss of NF1 Gene Activity

The neurofibromin 1 gene (*NF1*) encodes for the neurofibromin protein, which plays a fundamental role in the regulation of cell growth by acting as a suppressor of the RAS-MAPK pathway. Neurofibromin functions as GTPase-activating protein (GAP) for RAS, converting the active GTP-bound form of RAS to its inactive GDP-bound state, effectively controlling RAS activity. In melanoma, loss or mutation of NF1 impairs the ability of neurofibromin to deactivate RAS, leading to continuous RAS activation. This persistent activity drives the MAPK/ERK pathway independently of upstream regulators, such as BRAF. As a result, even under treatments with BRAFi, the pathway remains active due to the uncontrolled activity of RAS, which signals through alternative RAF proteins such as CRAF, allowing tumor cells to continue to proliferate.

In addition to resistance to BRAFi, loss of NF1 also confers resistance to MEKi, which targets MEK1/2 kinases downstream of BRAF in the MAPK pathway. Because the RAS remains active due to the absence of neurofibromin, it stimulates MEK activation even in the presence of MEKi, further reducing the effectiveness of such therapies.

Therefore, melanomas with NF1 deficiency are particularly difficult to treat with conventional inhibitors of the MAPK pathway. Loss-of-function mutations in NF1 occur in nearly 20% of melanomas, manifesting as missense or truncating mutations, as well as chromosomal deletions [10,34]. These alterations disrupt NF1’s ability to regulate RAS signaling, contributing to melanoma progression [33,35].

#### 6.1.5. RAC1 Mutation

RAC1 is a small GTPase from the Rho family, regulating crucial cellular processes like cell proliferation, cytoskeleton organization, and migration. The specific mutation RAC1 R29S, identified in up to 9% of sun-exposed melanomas, results in its constitutive activation promoting downstream signaling, driving cancer cell proliferation and survival, cytoskeleton remodeling, and invasion. The RAC1 P29S mutation is responsible for oncogenic resistance in melanoma treatment with BRAFi [27,36,37]. Nonetheless, larger cohort studies are still missing, and the significance of RAC1 mutations in melanoma and their impact on systemic treatment strategies remain unclear. Recently, Lodde and collaborators showed the largest group of RAC1-mutated melanoma patients reported so far (*n* = 64), with RAC1 mutation identified in approximately 2% of samples; the most frequent was PS9S; 84% were cutaneous melanoma, and 14% of unknown primary [38].

#### 6.1.6. Loss of the USP28-FBW7

The complex deubiquitinase enzyme USP28-ubiquitin ligase FBW7 plays a crucial role in regulating the stability of proteins, particularly those involved in cell signaling and cell cycle control. FBW7 is a ubiquitinating ligase that usually targets proteins for proteasomal degradation. However, USP28 acts as a deubiquitinase, removing ubiquitin chains from target proteins and preventing their degradation. The loss of this complex disrupts this balance, leading to uncontrolled degradation of several regulatory proteins.

In the context of melanoma, the loss of USP28-FBW7 complex has been identified as a significant mechanism to BRAFi in the treatment of metastatic melanoma: it causes the accumulation of oncogenic proteins, such as MYC, whose stability is normally regulated by FBW7. This accumulation may promote the reactivation of the MAPK pathway. Reactivation of this pathway allows tumor cells to evade BRAF inhibition, thus promoting proliferation. Loss of USP28-FBW7 also contributes to resistance by inhibiting the degradation of other proteins involved in cell survival, such as NOTCH1 (Figure 3). Furthermore, the accumulation of pro-oncogenic proteins accelerates the cell cycle, making tumor cells more resistant to targeted therapy [39].

### 6.2. Primary (Intrinsic) Resistance: Epigenetic and Transcriptomic Mechanisms

Epigenetic and transcriptomic mechanisms (Table 3 (a–c)) play critical roles in the development and progression of BRAF-mutant melanoma, significantly contributing to tumor heterogeneity, therapeutic resistance, and immune evasion. Epigenetic refers to non-genetic heritable mechanisms influencing gene expression without alterations in the DNA sequence [27]. These mechanisms generally include DNA methylation, mRNA processing and splicing, histone modifications, microRNAs, regulation of RNA stability, and chromatin remodeling. In the context of cancer, such alterations enable oncogenic mutations, facilitating tumorigenesis. Furthermore, these mechanisms contribute to both inter- and intratumoral heterogeneity, leading to therapeutic resistance and tumor progression [40].

In BRAF-mutant melanoma, tolerance to BRAFi and MEKi is closely linked to specific phenotypic plasticity, involving shifts in melanoma differentiation states and global changes in gene expression profiles [41]. Phenotype plasticity refers to an organism’s ability to adjust its phenotype in response to the local microenvironment, without altering its genome [42]. This plasticity includes transitions between various cellular states associated with melanocyte inducing transcription factor (MITF), a pivotal regulator of melanocyte lineage [43,44,45]. Additionally, melanoma cells can transition to a neural crest-like state, characterized by the expression of NGFR (nerve growth factor receptor) and other markers, which represent melanocyte precursors [46]. Another observed state is the undifferentiated phenotype, marked by low levels of SOX10 (SRY-related HMG-box 10) transcription factor and elevated expression of tyrosine kinases such as AXL and EGFR, associated with aggressive and drug-resistant melanoma subtypes [47].

Despite the multiple differentiation states, the individuals may vary from one tumor to another. Tumor heterogeneity affects the strength of signaling pathways through which each tumor responds and adapts to BRAF and MEKi [48]. The transcriptional heterogeneity and epigenetic reprogramming play a critical role in facilitating drug resistance in BRAF V600E melanoma cells. Indeed, prior to the drug addition, the reprogramming induces generation of heritable resistance phenotype due to the dynamic environment that allows for transition from non-resistant to pre-resistant phenotypes. This finally permits cell survival and leads to stable drug resistance [49,50].

It is evident that BRAF-mutant melanomas treated with MAPK inhibitors are able to adapt over time. During this adaptation process, alterations in the transcriptome or epigenome may be transient and reversible upon drug removal [46]. However, prolonged drug exposure can stabilize these transcriptionally and epigenetically altered phenotypic states [40]. As previously mentioned, epigenetic mechanisms that regulate gene expression and phenotypic states in melanoma may include DNA methylation, histone-modifying enzymes, and various histone modifications, which contribute to the development and maintenance of drug-resistance phenotypes.

#### 6.2.1. DNA Methylation

DNA methylation in eukaryotes provides methyl group (-CH_3_) covalent transfer to cytosine bases of DNA through DNA methyltransferases (DNMTs). DNMTs use S-adenosyl methionine (SAM) donor to transfer a methyl group to the 5-carbon of cytosine, producing 5-methylcytosine (5-mC) [51]. In particular, CpG sites are usually involved, regarding dinucleotides region made up by cytosine linked to a guanine base by a phosphodiester bond. These regions are usually methylated [52]. In melanoma, abnormal DNA methylation patterns are frequently observed and play an important role in its formation and progression [53]. Silencing of PTEN, p16/14, and RASSF1A (Ras association domain family 1, isoform A) have been investigated.

As mentioned above, *PTEN* is a tumor suppressor gene that inhibits the activation of the PI3K/AKT pathway [53]; its deletion or mutation occurs commonly with functional loss in about 40–60% of sporadic melanomas [52,53]. Furthermore, epigenetic *PTEN* silencing mediated by its promoter methylation has been reported in 62% of metastatic melanoma serum samples, contributing to the constitutive activation of the AKT pathway and melanoma development [54]. Thus, PTEN is correlated as a factor of poor prognosis and survival in melanoma [55].

p16^INK4A^ protein is encoded by the cyclin-dependent kinase inhibitor 2A (CDKN2A) locus and plays a pivotal role in arresting the cell cycle at G1/S checkpoint by inhibiting the cyclin D-dependent kinase 4/6 (CDK4; CDK6), while activating the phosphorylation of retinoblastoma protein (pRb) [56]. As a result, the protein leads to cell cycle arrest. The promoter methylation has been reported in 19% of primary cutaneous melanoma cases; the frequency is higher in metastases (33%), indicating that hypermethylation may occur also in later stages [57,58]. p14^ARF^, also encoded by the CDKN2A locus, binds and inhibits MDM2 from triggering p53 ubiquitination and targeting it for proteasomal degradation. As a consequence, a p53 protein level increase induces cell cycle arrest, allowing either DNA repair and cell survival or apoptosis [59]. The methylation of p14^ARF^ promoter has been observed in 57% of metastatic melanoma samples, independently of the p16^INK4A^ promoter [60].

The Ras-associated domain family 1 (*RASSF1*) gene contains eight exons and gives rise to eight different transcripts, from RASSF1A to RASSF1H. The RASSF1A tumor suppressor regulates cellular homeostasis such as RAS, MST2/Hippo, p53, and death receptor pathways. Moreover, it is involved in the inhibition of cell proliferation and activates intrinsic and extrinsic apoptotic cascades [61]. Methylation of two regions of RASSF1A CpG island was reported in 44 metastatic melanoma tumors and 11 melanoma cell lines [53]. Overall, RASSF1A was hypermethylated in 55% of melanoma tumors, which correlated with the loss of expression of the RASSF1A gene. RASSF1A was methylated in 15% (3/20) of primary tumors and 57% (49/86) of metastatic samples, suggesting that methylation level may correlate with advanced stage.

Other genes have been investigated for their methylation in melanoma, including Retinoic Acid Receptor beta 2 (*RARβ2*) [62], O6-MethylGuanine DNA Methyl Transferase (*MGMT*) and Death-Associated Protein Kinase (*DAPK*) [58,63]. All of them have been associated with hypermethylation in primary and metastatic melanomas while representing a key epigenetic factor in melanocyte transformation and tumor progression.

Hypomethylation, on the other hand, can activate normally silent oncogenes or repetitive DNA sequences, contributing to genomic instability. It has been observed that malignant tumor tissues are characterized by global DNA hypomethylation [64]. Indeed, demethylation has been identified as a putative hallmark of metastatic primary melanomas [65]. In particular, it has been observed that the hypomethylation of the TBC1D16 gene generates metastases in melanoma: TCB1D16–47KD isoform hypomethylation in metastatic patients was associated with a shorter PFS and OS. Notably, the unmethylated state was observed in 33% of patients carrying the BRAF V600E mutation and is associated with increased sensitivity to BRAFi [66].

#### 6.2.2. Histone Modifications

In eukaryotic cells, the nucleosome serves as the fundamental structural unit of chromatin, comprising DNA wrapped around an octamer of four core histone proteins. This octamer contains two copies each of core histones H2A, H2B, H3, and H4. A linker histone, H1, functions to stabilize the nucleosome structure. Histone proteins are subjected to various post-translational modifications, including acetylation, methylation, phosphorylation, and ubiquitination, which significantly expand the complexity of epigenetic regulation.

Among these modifications, the lysine methyltransferase SETDB1 has been implicated in melanoma progression. SETDB1 exerts its function by methylation histone H3 at lysine 9 (H3K9), a modification linked to transcriptional repression, and its overexpression has been associated with increased melanoma formation. Conversely, H3K9 lysine demethylases, including LSD1 and JMJD2C, have been shown to cooperate with mutant BRAF to overcome oncogene-induced senescence (OIS), a cellular mechanism that normally suppresses tumorigenesis. By bypassing OIS, these enzymes facilitate the development of malignant melanoma [67,68]. This highlights the role of epigenetic reprogramming in enabling BRAF-mutated cells to exploit OIS escape pathways, promoting tumor formation and progression. Furthermore, exposure of BRAF V600E melanoma cells to sublethal doses of BRAF and MEKi induces a shift toward a drug-tolerant state. This state is characterized by elevated expression of histone demethylases such as KDM6A, KDM6B, KDM1B, and JARID1A, along with decreased levels of histone marks H3K4me3 and H3K27me3 [69,70]. These alterations reflect the epigenetic activation of certain genes and the silencing of others, facilitating cellular adaptations that confer survival advantages in the presence of MAPK pathway inhibitors [70]. The histone deacetylase SIRT6 was found to be downregulated in BRAF V600E melanoma cells resistant to MAPK inhibitors. This downregulation was identified as a key factor contributing to resistance against BRAF and MEKi [71].

Overall, these findings underline the role of histone-modifying enzymes in mediating epigenetic changes that influence drug resistance, emphasizing the potential of targeting these mechanisms to overcome therapeutic challenges in melanoma.

#### 6.2.3. Transcriptome Regulation

Melanoma is characterized by a highly mutagenized genome, resulting in a substantial neoantigen load that makes it more immunogenic compared to other malignancies. Transcriptional regulation, a key determinant of transcriptome and proteome diversity, is frequently disrupted in melanoma due to genetic and epigenetic alterations. These disruptions exploit transcriptional networks, inhibit tumor-suppressive transcription, and enhance oncogenic transcription factor activity, driving pro-oncogenic gene expression signatures. Understanding how transcriptional network dysregulation influences melanoma cell proliferation and survival is critical for comprehending disease progression and therapeutic resistance [72,73]. Transcriptome-wide profiling of patient-derived melanoma samples has revealed specific signatures reflective of melanoma subtypes. In general, the transcription of coding RNAs (mRNAs), noncoding RNAs (microRNAs, miRNAs), long noncoding RNAs (lncRNAs), transfer RNAs (tRNAs), and ribosomal RNAs (rRNAs)) is downregulated in melanoma [74,75].

An example of melanoma-driving transcription factor is the melanocyte inducing transcription factor (MITF), overexpressed in 20% of metastatic melanomas and necessary for the survival of normal human melanocytes [76]. The recurrent E318K point mutation in MITF was identified and associated with gain-of-function mutation and associated with familial and sporadic melanoma [77]. In some cases, MITF can be post-translationally modified by sumoylation, resulting in a reduced transcriptional activity [78]. MITF E318K mutation abolishes prevent the protein sumoylation, a post-translational modification involving the conjugation of small ubiquitin-like modifiers (SUMOs) to lysine residues or target proteins, resulting in an enhanced MITF transcriptional activity and MITF-regulated gene expression [77]. Additionally, MITF activity is modulated by phosphorylation at specific residues by a number of kinases such as c-KIT, who upregulates MITF expression via MEK/ERK pathway-mediated phosphorylation at Ser73 residue [78]. Similarly, GSK3β and p38 mitogen-activated protein kinase (MAPK) have shown to phosphorylate MITF at Ser298 and Ser307 residues [79].

Another interesting example with a pivotal role in melanoma is the transcription factor SOX10, essential for neural crest and oligodendrocyte development [80]. In melanoma, SOX10 is phosphorylated and regulated by ERK-mediated sumoylation at Lys55, a modification crucial for its transcriptional activity and target gene selection, particularly in BRAF-mutant melanoma [81]. This highlights SOX10’s role in both the initiation and progression of the disease.

Of note, the tumor suppressor gene p53 is also frequently mutated in melanoma, with alterations occurring in approximately 15% of cases [82]. Typically, these mutations are either in-frame or missense and prevent p53 from binding to DNA, thereby impairing the ability of p53 to activate its target genes [83]. Furthermore, some mutations result in a gain of function, enabling p53 to acquire oncogenic properties that promote tumor growth and survival [84].

Transcriptomic analyses have demonstrated that the persistent activation of the MAPK/ERK signaling pathway in BRAF V600E mutation reprograms cellular gene expression, promoting oncogenic transcriptional profiles. A defining feature of BRAF-mutant melanoma is the upregulation of proliferation-associated genes, such as *MYC*, CCND1, and E2F1, which synergistically fuel the unchecked growth of melanoma cells [85,86]. In parallel, anti-apoptotic genes, including *BCL2* and *MCL1*, are commonly overexpressed, conferring resistance to programmed cell death and enhancing tumor cell survival under therapeutic pressure [87,88,89].

Finally, miRNAs play a pivotal role in melanoma biology by modulating post-transcriptional gene expression. For example, miR-211, miR-205, and miR-200c, which are often downregulated in BRAF-mutant melanoma, normally act as tumor suppressors by targeting oncogenic pathways [90,91,92]. Conversely, upregulated miRNAs, such as miR-10b, promote metastasis by repressing anti-invasive factors [93,94]. Collectively, these studies underline the intricate interplay between epigenetic and transcriptomic regulation in melanoma.

### 6.3. Secondary Resistance

Secondary resistance to BRAF inhibitors in melanoma (Table 4) can be classified into two forms: adaptive resistance (Figure 4), which arises early through reversible cellular reprogramming, and acquired resistance (Figure 5), which develops later through stable genetic or epigenetic alterations.

#### 6.3.1. Adaptive Resistance

Adaptive resistance is characterized by an initial response to BRAFi treatment, which diminishes over the course of continued therapy (Figure 4). This form of resistance gradually develops mechanisms to evade treatment without involving genetic mutations. The mechanisms that drive adaptive resistance in melanoma are primarily composed by resetting the activity of the ERK1/2 pathway, upregulation of receptor tyrosine kinases and upregulation of MIcrophthalmia-associated Transcription Factor (MITF).

##### Reactivation of the ERK1/2 Pathway

The main cause of the adaptive resistance to targeted therapy is represented by the reactivation of the ERK1/2 signaling, which could be triggered by the impingement of two negative feedback mechanisms for the MAPK pathway, involving the Sprouty proteins SPRY2 and SPRY4 or the phosphatase DUSP4 (dual specificity phosphatase 4) and DUSP6 (dual specificity phosphatase 6). The first proteins inhibit the receptor tyrosine kinase signaling, including the RAS-RAF-MEK-ERK pathway, by interfering with RAS activation, while the latter are dual-specificity phosphatases that deactivate ERK by removing phosphate groups. During treatment with BRAFi, this negative feedback is reduced with the consequent release of RAS and other proteins upstream of the MAPK pathway; however, a diminution of SPRY2, SPRY4, DUSP4, and DUSP6 expression was observed. In BRAF V600E mutant cells, RAS activity is low because the BRAF mutation is sufficient to keep the MAPK pathway active [95,96].

Recent findings have identified p38 MAPK as a novel mediator in the adaptive response of melanoma cells to BRAF-targeted therapy. This study demonstrated that treatment with BRAFi triggers an early increase in p38 activation, which, in turn, promotes the phosphorylation of the transcription factor SOX2 at Ser251. This phosphorylation enhances SOX2’s stability, its localization to the nucleus, and its transcriptional activity, contributing to the survival of melanoma cells under BRAFi treatment. Further functional studies revealed that depletion of SOX2 heightened the sensitivity of melanoma cells to BRAFi, while overexpression of a phosphomimetic SOX2-S251E mutant (mimicking constant phosphorylation at Ser251) was sufficient to drive resistance. This overexpression desensitized melanoma cells to treatment both in vitro and in a zebrafish xenograft model, reinforcing the role of SOX2 in mediating resistance. Additionally, phosphorylation of SOX2 at Ser251 was found to increase resistance to BRAFi by upregulating the transcription of the *ABCG2* gene, which encodes an ATP-binding cassette drug efflux transporter. This transporter helps melanoma cells expel therapeutic agents, contributing further to the drug resistance phenotype [97].

##### Receptor Tyrosine Kinases (RTKs)

RTKs are transmembrane proteins that respond to extracellular signals, such as growth factors, to activate intracellular signaling cascades, including the MAPK and PI3K/AKT/mTOR pathways. Recent research has identified transcriptional activity as a possible mechanism contributing to adaptive resistance to BRAFi [98].

In BRAF-mutated melanoma cells, the inhibition of ERK1/2 signaling leads to the induction of forkhead box D3 (FOXD3), which influences the sumoylation of SOX10. This, in turn, enhances the transcription of ERBB-3 (HER3), a receptor tyrosine kinase from the EGFR family. However, ERBB-3 lacks critical catalytic residues, including the aspartate catalytic base, and requires dimerization with other ERBB family members to function. ERBB-2 is the preferred partner for dimerization across the ERBB family, and ERBB-3 tends to pair with ERBB-2 [99]. Multiple studies have reported that ERBB-3 becomes phosphorylated when BRAF V600E-mutated melanoma cells are treated with BRAFi or MEKi. Specifically, the phosphorylation at Y1289 on ERBB-3 creates a binding site for PI3K, leading to the activation of the PI3K/AKT/mTOR pathway [100]. Furthermore, increased ERBB-3 expression leads to the upregulation of neuregulin 1 (NRG1), which preferentially binds to ERBB-2 and enhances its phosphorylation on tyrosine residues. On the other hand, NRG1 is highly expressed by dermal fibroblasts and cancer-associated fibroblasts in melanomas with BRAF mutations, promoting cell growth and resistance to BRAFi [101].

In a recent study, the SOX10/MITF axis was investigated in relation to the RTK axis in vemurafenib resistance. The authors analyzed the gene expression levels of ligands and receptors previously associated with the SOX10/MITF axis in melanoma cell lines. The study found that SOX10, MITF, ERBB3, and GAS6 were upregulated during the early stages of treatment with BRAFi, while EGFR, AXL, and NRG1 were upregulated after the development of resistance to vemurafenib [102].

##### MIcrophthalmia-Associated Transcription Factor (MITF)

MITF is a transcription factor specific to the melanocytic lineage, which plays a significantly expanded role in malignant melanoma. In recent years, MITF has been closely linked to the plasticity of melanoma cells, contributing to various melanoma phenotypes characterized by distinct gene expression profiles [103]. When BRAF or MEK is inhibited, the feedback mechanisms in melanoma cells trigger compensatory pathways to reactivate or increase MITF expression. Despite the suppression of MAPK signaling by inhibitors, some transcriptional regulators (such as SOX10) and other factors can still activate MITF expression through MAPK-dependent transcriptional circuits. This allows melanoma cells to maintain growth and resist the effects of BRAF/MEKi.

When MITF is upregulated in response to BRAF/MEK inhibition, it promotes cell survival, differentiation, and metabolic adaptations that enable melanoma cells to escape drug-induced cell death. MITF can also induce the expression of survival genes and help melanoma cells switching to alternative growth pathways, further enhancing resistance to therapy. This adaptive response allows melanoma cells to survive and continue proliferating despite the initial efficacy of BRAFi or MEKi [104].

##### SOX10

The Sex-determining region Y-bOX 10 (SOX10) belongs to the SOX family of transcription factors, whose role is crucial in the development of the neural crest and melanocyte lineage. SOX10 promotes the proliferation and growth of melanoma, activating specific targets such as MIT, long non-coding RNA (Inc RNA), SAMMSON, and FOXD3. In turn, SOX10 expression is under the control of regulatory sequences in the region-encoding gene that can be bound by other transcriptional factors. Recently, it was shown that a loss of SOX10 expression caused vemurafenib resistance through the activation of transforming growth factor beta (TGFβ) signaling, leading to the upregulation of epithelial growth factor receptor (EGFR) and platelet-derived growth factor receptor (PDGFR). However, it has been shown that the reducing expression of SOX10 or the depletion of its target SAMMSON increases the sensitivity of BRAF-mutant melanoma to MAPK inhibitors, suggesting that upregulation of SOX10 promotes resistance to inhibitor therapy. The paradoxical role of SOX10 in resistance to BRAFi depends on the specific phase of therapy. During the early phase, SOX10 activity increases, promoting adaptive resistance through the upregulation of cytoprotective factors. In contrast, during the later phase associated with acquired resistance SOX10 expression is downregulated. This reduction leads to the compensatory reactivation of RTKs, which supports tumor cell survival rather than inhibiting it [105].

#### 6.3.2. Acquired Resistance

Mechanisms of acquired resistance to BRAFi in melanoma are complex and multifaceted (Figure 5). These mechanisms enable melanoma cells to bypass the targeted inhibition of the MAPK pathway (RAS-RAF-MEK-ERK), allowing for continued proliferation and survival despite treatment. It usually develops after an initial period of tumor shrinkage or stabilization, leading to eventual disease progression. The key mechanisms driving this resistance are NRAS mutations, the RAF paradox and dimerization of RAF proteins, *BRAF* gene amplification and splicing variants, MEK1/2 mutations, and the tyrosine kinase receptor hyperactivation [106,107].

##### NLRP1 in MAPK/ERK Pathway

The constant activation of the MAPK/ERK pathway triggers various downstream effector responses. One such effector is the inflammasome sensor NLRP1 (NACHT, LRR, and PYD domains-containing protein 1), which promotes melanoma growth by facilitating IL-1β maturation. In a study, NLRP1 operates as a downstream effector of MAPK/ERK signaling via activating transcription factor 4 (ATF4) in metastatic melanoma cells, where ATF4 directly regulates NLRP1. The regulation of ATF4/NLRP1 by the MAPK/ERK pathway changes after the cells resist targeted therapies.

In parental cells, this regulation occurs through ribosomal S6 kinase 2 (RSK2). However, in cells resistant to vemurafenib and trametinib, the regulation shifts to the cAMP/protein kinase A (PKA) pathway. Consequently, while NLRP1 expression and IL-1β secretion are reduced in response to vemurafenib and trametinib in parental cells, they are elevated in resistant ones. Notably, silencing NLRP1 in resistant cells significantly hindered their growth and colony formation. These findings suggest that NLRP1, regulated through ATF4, is a key player in resistance to targeted therapies in melanoma, and its inhibition could enhance the therapeutic response [108].

##### NRAS Mutations

Acquired resistance is frequently driven by NRAS mutations, which were first identified by Nazarian and colleagues in 2010. These mutations occur in approximately 20% of melanomas and are often associated with a more aggressive tumor phenotype and reduced patient survival [109]. To better understand the mechanisms underlying this resistance, researchers created BRAF V600E melanoma clones with acquired resistance to vemurafenib through chronic selection. These resistant clones retained the V600E mutation, and no additional mutations were found in the BRAF coding sequence. Further analysis revealed that vemurafenib failed to inhibit ERK phosphorylation, indicating reactivation of the pathway [109].

Resistance has also been associated with elevated RAS-GTP levels, and sequencing of *RAS* genes identified a rare activating mutation in KRAS, resulting in a K117N substitution. Furthermore, combined treatment with vemurafenib and a MEKi or an AKT inhibitor synergistically suppressed the proliferation of resistant cells [110]. These findings suggest that resistance to BRAF V600E inhibition may occur through multiple mechanisms, including elevated RAS-GTP levels and increased AKT phosphorylation. Data indicate that reactivation of the RAS/RAF pathway via upstream signaling is a key mechanism of acquired resistance to vemurafenib [110].

In a recent study a researcher performed single-cell RNA sequencing on NRAS-mutant melanomas treated with MEK1/2 and CDK4/6 inhibitors to decode the transcriptional transitions occurring during the development of drug resistance. They identified cell lines that resumed full proliferation (FAC, fast-adapting cells) and others that became senescent (SAC, slow-adapting cells) after prolonged treatment [111]. The initial drug response was characterized by transitional states involving increased ion signaling, driven by the upregulation of the ATP-dependent ion channel P2RX7. The activation of P2RX7 was associated with improved therapeutic responses, and when combined with targeted therapies, it could help delay the onset of acquired resistance in NRAS-mutant melanoma [111].

##### BRAF Paradox and RAF Proteins Dimerization

PLX47206, sorafenib, and dabrafenib have been developed as inhibitors that effectively block the MAPK signaling pathway, reducing tumor growth in cells harboring the BRAF V600E mutation. While these inhibitors successfully suppress ERK signaling in BRAF V600E mutant cells, they exhibit unexpected agonistic effects in cells with wild-type BRAF. Specifically, when these inhibitors bind to the wild-type BRAF monomer, they promote homo- or heterodimerization with another RAF protomer (such as BRAF/BRAF or BRAF/CRAF) in the presence of KRAS loaded with GTP at the membrane.

Recent studies have shown that dimerized BRAF has a lower affinity for the inhibitors compared to ATP, yet it retains high kinase activity, continuing to activate the downstream MAPK pathway. Since RAS-GTP levels are significantly lower in BRAF V600E cells, dimer-mediated activation of the MAPK pathway is less likely in these cells. However, in cells expressing wild-type BRAF, the binding of these inhibitors paradoxically activates the wild-type BRAF kinase domain by inducing conformational changes that lead to dimerization and CRAF activation, ultimately resulting in MEK/ERK phosphorylation and increased cell growth.

Therefore, ATP-competitive inhibitors can either block or stimulate the MAPK pathway, depending on whether the tumor cells express wild-type BRAF or the BRAF V600E mutation [112]. In a recent study it emerged that a novel imidazothiazole-based compound, KS28, is capable of overcoming PLX4032 resistance in melanoma by targeting and downregulating the MEK/ERK signaling pathway. The researcher investigated the effect of KS28 on PLX4032-resistant melanoma cells and observed that this compound can inhibit RAF dimerization leading to a suppression of the MEK/ERK pathway, responsible for cell survival and proliferation. In addition to blocking the pathway, KS28 reduced the activation of activator protein 1 (AP-1) in resistant melanoma cells, decreased cell viability, and promoted DNA fragmentation. Further experiments demonstrated that KS28 could inhibit the anchorage-independent growth of A375R cells, a key feature of cancer progression [113].

##### BRAF Gene Amplification and Splicing Variants

*BRAF* gene amplification and splicing variants represent key mechanisms of acquired resistance to BRAFi in melanoma treatment. *BRAF* gene amplification occurs when melanoma cells increase the number of *BRAF* gene copies, leading to an overexpression of the mutant BRAF V600E protein. Normally, BRAFi bind to this mutant protein, blocking its activity and suppressing the MAPK pathway. However, when BRAF is amplified, the excess production of BRAF V600E overwhelms the inhibitor, resulting in just a partial inhibition. This leaves some BRAF molecules active, allowing the MAPK pathway to continue functioning and promoting tumor cell growth despite the inhibitor’s presence.

In addition to gene amplification, resistance also arises through the generation of *BRAF* splicing variants. These variants, produced through alternative splicing of the *BRAF* gene, lack the RAS-binding domain required for normal BRAF function. While BRAF typically forms dimers (pairs) with RAS to activate the MAPK pathway, these splicing variants can form dimers independently of RAS, creating BRAF-BRAF or BRAF-CRAF dimers. These dimers are resistant to BRAFi, which target only monomeric BRAF. As a result, the MAPK pathway is reactivated, allowing tumor cells to survive and proliferate, even in the presence of BRAFi. Recently a new genomic alteration about inhibitor BRAF resistance in melanoma emerged. Indeed, a study by Aya et al. [114] highlighted that alternative BRAF mRNA isoforms (altBRAFs), observed in about 30% undergoing BRAFi therapy, were the products of alternative splicing but were effectively generated through genomic deletions. In particular, the authors showed, in various in vitro models of altBRAF-driven melanoma resistance, that altBRAFs are produced exclusively from the BRAF V600E allele and are associated with corresponding genomic deletions. Moreover, the presence of these deletions in BRAF wild type model melanoma samples suggested a significant shift in understanding the mechanisms behind the generation of BRAF transcript variants that contribute to resistance in melanoma [114].

To study the mechanisms of acquired drug resistance, clones resistant to the MEKi AZD6244 were developed from two colorectal cancer cell lines with the BRAF V600E mutation. These clones also showed cross-resistance to BRAFi, due to amplification of the *BRAF* gene. A small group of parental cells already had pre-existing amplification of *BRAF*, which was also found in some BRAF-mutated colorectal cancer cells. BRAF amplification increased levels of phosphorylated MEK, reducing the effectiveness of AZD6244 in inhibiting ERK phosphorylation. However, ERK inhibition was restored by treating the cells with low doses of BRAFi. The combination of MEK plus BRAFi completely reversed resistance and was more effective than either treatment alone, indicating that BRAF amplification is a key mechanism of resistance [115].

##### MEK1/2

Acquired resistance in melanoma is also linked to secondary mutations in MEK1 and MEK2, which are present in approximately 7% of melanomas that have developed resistance to BRAFi. In a study, researchers explored the significance of MEK1 mutations in BRAFi resistance by parallel sequencing on resistant clones derived from a MEK1 random mutagenesis screen in vitro, as well as tumors from patients who relapsed after treatment with AZD6244, an allosteric MEKi. The authors observed that the mutations that confer resistance to MEKi are primarily found in the allosteric drug binding pocket or α-helix C, leading to a significant resistance to MEK inhibition. Furthermore, other mutations, including MEK1(P124L) and MEK1(Q56P), located near the N-terminal regulatory helix induced cross-resistance to the BRAFi PLX4720 in melanoma [116]. De novo MEK2-Q60P mutation was observed in melanoma from patients treated with the MEKi trametinib not responding to the BRAFi dabrafenib. The same mutation was also found in another patient resistant to the combination of dabrafenib and trametinib. These resistant cells exhibited continued MAPK activation and persistent S6K phosphorylation. A triple therapy combining dabrafenib, trametinib, and the PI3K/mTOR inhibitor GSK2126458 effectively inhibited tumor growth [115,117].

##### The Tyrosine Kinase Receptor (RTK)

In many cancer types, RTKs can undergo hyperactivation by some mechanisms, such as overexpression, mutations, or stimulation by ligands, causing an uncontrolled tumor growth. In 2017, a large sequencing throughout the genome (WGS) on melanoma highlighted that the alterations of the receptor tyrosine kinase pathway were present in 42% of melanoma cases. The overexpression of RTK is responsible for the BRAFi resistance in melanoma treatment, activating the parallel or direct RAS pathways [118].

The RTK hyperactivation in melanoma is mainly due to EGFR, PDGFRβ, and IGF1R receptor overexpression or genetic mutations constantly activating RTK. In a study by Nazarian and collaborators, it emerged that the upregulation of PDGFRβ causes the resistance to PLX403 [109]. Using PLX4032-resistant cell sublines, artificially derived from BRAF V600E-positive melanoma cell lines, key findings were validated in PLX4032-resistant tumors and in short-term cultures derived from patients involved in clinical trials. Tumor cells with upregulated PDGFRβ showed low levels of RAS activity and, after the treatment with PLX4032, were unable to activate the MAPK pathway again. Then, melanoma eludes BRAF V600E targeting by the reactivation of alternative survival pathways through RTK or MAPK pathway reactivation mediated by activated RAS. Also, EGFR and IGF1R are constantly activated in melanoma cells’ resistance. In particular, a study showed that the downregulation of SOX10 in melanoma activated TGFβ to induce upregulation of EGFR and PDGFRβ, which then conferred resistance to MAPK inhibition. Furthermore, it was shown that IGFR1 signaling greatly enhanced the activation of the PI3K/AKT pathway in resistant cells [109].

##### PI3K/AKT Pathway

Reduced ERK signaling in melanoma cells may cause hyperactivation of the PI3K/AKT/mTOR pathway, contributing to the development of acquired resistance to BRAFi. Cells with high PI3K/AKT pathway activity gain a survival advantage by becoming less sensitive to the effects of drugs that target BRAF. This enhancement of the PI3K/AKT pathway can occur through various mechanisms, including increased expression of the IGF1R receptor, which promotes persistent signaling capable of inhibiting apoptosis and promoting cell survival. Mutations that activate the PI3K/AKT pathway have been observed in approximately 22% of melanoma patients who developed treatment resistance. These mutations increase AKT activity, leading to the activation of anti-apoptotic signals and pro-proliferative genes, allowing tumor cells to grow independently of BRAF signaling and resist MAPK pathway inhibition [119].

##### STAG2 or STAG3 Expression and YAP/TAZ Pathway

Another important mechanism contributing to acquired resistance is the reduction of STAG2 and STAG3 protein expression, found in patients resistant to BRAFi. The loss of these proteins reduces the response of melanoma cells to MAPK pathway inhibition and results in the reactivation of ERK signaling through reduced activity of the DUSP6 phosphatase, responsible for ERK regulation. Furthermore, the YAP/TAZ pathway part of the Hippo signaling pathway has an oncogenic role by contributing to the remodeling of the actin cytoskeleton in resistant melanoma cells. This remodeling enhances the nuclear translocation and transcriptional activity of YAP and TAZ, stimulating the expression of molecules that regulate the cell cycle and facilitating resistance. Inhibiting YAP or TAZ can restore the effectiveness of BRAFi by preventing the constant activation of ERK1/2 [120,121].

##### The Expression of Dual-Specificity MAPK Phosphatases (DUSPs)

Another important component is represented by dual-specificity MAPK phosphatases (DUSPs), such as DUSP4 and DUSP6, which are responsible for the dephosphorylation and inactivation of ERK. Recently it was shown that DUSP proteins are involved in cell regulation of cell plasticity and drug resistance of melanoma and are potential targets for the treatment of MAPK inhibitor resistant melanoma. Indeed, the authors observed that silencing DUSP1 or DUSP8, or treatment with BCI, a pharmacological inhibitor of DUSP1/6, reduces the survival of MAPKi-resistant cells and sensitizes them to BRAFi and MEKi therapies. Pharmacological inhibition of the upregulated DUSP1/6, a marker of neural crest stem cells such as nestin, was effective in both MAPKi-sensitive cells and those with acquired resistance to MAPKi. Interestingly, BCI treatment led to the upregulation of MAP2, a neuronal differentiation marker, only in MAPKi-sensitive cells, while it caused downregulation of both MAP2 and GFAP, a glial marker, across all MAPKi-resistant cell lines. The expression of these phosphatases is directly regulated by oncogenic BRAF mutations. BRAF-mutant melanoma cells show higher levels of DUSP6, and its reduction after treatment with BRAFi suggests that loss of negative feedback control over ERK contributes to resistance [122].

##### The Expression of Ring Finger Protein 125 (RNF125)

Another mechanism contributing to acquired resistance is the decreased RNF125 protein expression in resistant melanoma cells. Reduced expression of RNF125 inhibits JAK1 ubiquitination and degradation, promoting EGFR activation and ERK pathway reactivation. Targeting JAK1 and EGFR signaling in cells with low RNF125 expression could represent an effective therapeutic approach to overcome resistance to BRAFi. In melanoma cell cultures and tumor samples, SOX10/MITF expression was found to correlate with and drive RNF125 transcription. Reduced RNF125 levels were linked to increased expression of receptor tyrosine kinases, such as EGFR. Importantly, RNF125 influenced RTK expression by regulating JAK1, which was identified as an RNF125 substrate. RNF125 bound to JAK1, promoting its ubiquitination and degradation, which in turn suppressed RTK expression. Targeting JAK1 and EGFR signaling successfully overcame BRAFi resistance in melanomas with low RNF125 expression, both in vitro and in in vivo xenograft models. These findings suggest that combination therapies inhibiting JAK1 and EGFR may be an effective strategy for treating BRAFi resistant tumors with low RNF125 expression [123].

### 6.4. Immune Mechanisms

The immune microenvironment plays a pivotal role in determining the response to BRAF inhibition, influencing both the initial therapeutic response and the development of resistance (Table 5). Indeed, BRAF inhibition is able to alter the tumor-immune interactions, leading to the formation of significant barriers to reach therapeutic success [124,125,126]. Inhibitors targeting the MAPK pathway contribute to the creation of a more favorable immune milieu through the suppression of immunosuppressive factors. These include interleukin-10 (IL-10), vascular endothelial growth factor (VEGF), programmed death-ligand 1 (PD-L1), myeloid-derived suppressor cells (MDSCs), regulatory T cells (Tregs), and inhibitory stromal fibroblast interactions [127,128,129,130]. Notably, Sumimoto et al. in 2006 firstly reported and identified the role of the MAPK pathway in immune evasion even before the advent of pharmacological inhibitors targeting the oncogenic BRAF mutations [129]. Their findings revealed that patients treated with MEKi against BRAF V600E exhibited a reduced expression in IL-6, IL-10, and VEGF, highlighting the critical role of the immunosuppressive axis mediated by this pathway [129]. Clinical studies on patients with metastatic melanoma treated with BRAFi, as monotherapy or in combination with MEKi, caused a decrease in intratumoral IL-6 and IL-8 levels. This was accompanied by an increase in exhaustion markers such as PD-1, PD-L1, and TIM-3 [125].

In addition, MAPK pathway inhibition exerts a dual effect on T cells, promoting intratumoral infiltration and enhancing T cell functional activity, related to resistance mechanisms presentation. This is achieved through increased melanoma differentiation antigen (MDA) expression via transcriptional repression and block of the pathway, including gp100, Trp-1, Trp-2, and MART-1 [131]. These changes stimulate antigen-specific T-cell responses and potentiate cytotoxic activity against tumor cells, affecting MHC-I internalization normally used by activated T cells, thereby improving antigen presentation on the tumor cell surface [130]. It has been reported that T-cell proliferation changed according to the targeted therapy: while BRAF inhibition did not show any effect in T-cell proliferative response, MEK inhibition led to decreased proliferation. BRAF inhibition causes T-cell activation in a dose-dependent manner, measured by upregulation of activation markers CD69 and Ki67; this corresponds to greater magnitude of ERK signaling [131]. Wilmott et al. demonstrated an increase in CD8+ and CD4+ T cells in patient tumor samples collected one week after treatment with BRAF inhibition, with magnitude of CD8+ T-cell influx correlating with tumor necrosis and size reduction [132]. Moreover, Cooper et al. demonstrated that BRAF inhibition leads to a higher percentage of activated intratumoral CD8+ T cells that secrete IFN-γ and TNF-α in a BRAF-mutant murine melanoma model [133].

Beyond T cells, BRAF/MAPK pathway blockade influences dendritic cells (DCs). These are antigen-presenting cells acting as a bridge between the innate and adaptive immune system, critical for effective activation of T-cell response. Oncogenic BRAF activity has been linked to impaired DC maturation, contributing to an immunosuppressive microenvironment [129]. Immunosuppressive cytokines in the tumor microenvironment were shown to mediate decreased production of IL-12 and TNF-α and decreased expression of the costimulatory markers CD80, CD83, and CD86 [134]. In addition to the effects on T cells and DCs, BRAF/MAPK pathway inhibitors may also modulate natural killer (NK) cell activity. NK cells are cytotoxic lymphocytes that are traditionally part of the innate immune system. These, in response to BRAF inhibition, can potentiate inhibitor response through a perforin-dependent pathway in a BRAF-mutant melanoma model [135].

Furthermore, BRAF/MAPK pathway blockade is represented by inhibition of tumor-associated fibroblasts (TAFs) by modulation of interleukin-1 (IL-1), in particular by the transcription of IL-1α and IL-1β [136]. The chemokine *CCL2*, known to be involved in tumor progression and metastasis, has been studied as well, demonstrating that BRAF-targeted therapy decreases *CCL2* gene expression and secretion, correlated with reduction in tumor size in a specific murine model of BRAF-mutated melanoma [137].

An emerging focus in the context of BRAFi therapy regards the role of tumor-associated macrophages (TAMs) in the setting of therapeutic outcomes. Macrophages are highly plastic immune cells, able to adopt distinct functional phenotypes in response to microenvironmental changes. It is known that macrophages can be polarized as M1 or M2 [138]. The M1 type is classically activated, plays a role in T helper (Th1) cellular immune response, and is thought to be pro-inflammatory, while the M2 type is alternatively activated, plays a role in Th2 immune responses, and is thought to be immunosuppressive and pro-tumoral [139]. Preclinical studies on murine BRAF mutant models treated with either BRAFi alone or BRAF/MEKi in combination led to an increase in the number of intratumoral macrophages, inhibiting the efficacy of targeted therapy [124].

It is clear that the tumor immune microenvironment is a critical factor influencing both the initial therapeutic response to BRAFi and the emergence of treatment resistance [125]. Therefore, any resistance mechanism that develops in the various components of the immune microenvironment in the context of BRAF/MAPK arising from the immune components within the tumor microenvironment can compromise the therapeutic benefit, necessitating further investigation [124,140]. One key mechanism investigated regarded the role of immunosuppressive ligands for PD-1, named PD-L1, in melanoma exposed to BRAFi. Research has shown that PD-L1 expression is significantly upregulated in BRAFi-resistant melanoma cells compared to parental lines. This increase is linked to enhanced MAPK pathway activity, since it has been proved that PD-L1 expression remains high for several months even without BRAFi [138]. Suppression of MAPK pathway via BRAFi, MEKi and ERK 1/2 siRNA effectively reduces PD-L1 expression, with addictive effects resulting from combination of BRAF and MEK inhibition.

These findings collectively highlight the intricate interplay between immune components and MAPK signaling in mediating resistance. Therapeutic strategies aimed at modulating the immune microenvironment, particularly macrophage activity and cytokine production, hold promise for mitigating resistance and improving patient outcomes in melanoma treated with BRAF/MAPK-targeted therapies.

### 6.5. Metabolic Pathway

Melanoma cells with BRAF mutation exhibit a significant alteration in their energy metabolism (Table 6), particularly through an increase in aerobic glycolysis (Warburg effect) [141]. Unlike the metabolism of normal cells, which primarily use oxidative phosphorylation in mitochondria to produce energy in the presence of oxygen, tumor cells, including those with BRAF mutation, favor glycolysis even under aerobic conditions. Aerobic glycolysis is less efficient than oxidative phosphorylation, as it produces only 2 molecules of ATP per glucose molecule, compared to about 36–38 molecules of ATP produced by oxidative phosphorylation. However, glycolysis is a faster process, allowing tumor cells to proliferate rapidly. Furthermore, in melanoma, BRAF plays a crucial role in this metabolic reprogramming due to its significant effect on glucose metabolism through various mechanisms. Activated BRAF induces metabolic reorganization by suppressing the oxidative phosphorylation gene program by reducing the expression of MITF and the mitochondrial master regulator PGC1.

Notably, in a subgroup of melanomas with relatively high MITF expression, PGC1 expression is consistently upregulated and correlated with increased mitochondrial biogenesis and oxidative phosphorylation. Reduction of MITF may lead to impaired transcription of PGC1 and therefore to suppression of transcription of mitochondrial oxidative metabolism genes, reducing mitochondrial function and compensatory activating glycolysis.

Melanoma cells with BRAF mutation not only increase glycolytic activity but also modify how they absorb and utilize glucose. Tumor cells upregulate the expression of glucose transporter type 1 (GLUT1) on their surface. GLUT1 is responsible for transporting glucose across the cell membrane, thus increasing glucose uptake by tumor cells to meet their high energy and biosynthetic needs. This enzyme catalyzes the first reaction of glycolysis, phosphorylating glucose to form glucose-6-phosphate, thereby trapping glucose inside the cell and marking the start of the glycolytic process. At the other end of glycolysis, pyruvate kinase (PK) catalyzes the final step, converting phosphoenolpyruvate (PEP) into pyruvate and generating ATP. The upregulation of PK in melanoma cells helps maintain a high flux of glycolytic intermediates, ensuring continuous energy production. Additionally, lactate dehydrogenase (LDH) is often upregulated in tumors, including melanoma, where it converts pyruvate into lactate. This process regenerates NAD+, which is critical for sustaining glycolysis, particularly under aerobic conditions (Warburg effect). This metabolic reprogramming supports the growth and survival of melanoma cells [141,142]. In summary, mutated BRAF in melanoma not only activates signaling pathways that promote cell proliferation but also induces complex metabolic reprogramming that supports the high biosynthetic demands of tumor cells.

This MITF-PGC1 axis connects BRAF mutation to mitochondrial dysfunction in melanoma. Additionally, BRAF mutations activate glycolysis by enhancing the expression of hypoxia-inducible factor 1a (HIF1a) target molecules, which are involved in glucose utilization and uptake. Recent phosphoproteomic studies have identified PFKFB2 as a downstream phosphorylation substrate of RSK in BRAF-mutant melanoma cells. PFKFB2 regulates the generation of fructose-2,6-bisphosphate, enhancing PFK-1 activity, a key enzyme in glycolysis. These findings were reinforced by observations that the knockdown of PGC1induces similar metabolic reprogramming as seen in MITF deficiency in melanoma, and knocking down PFKFB2 significantly reduces glycolytic capacity and leads to tumor regression.

RSK directly phosphorylates PFKFB2, promoting glycolysis and melanoma development [143,144]. A recent study investigated metabolic reprogramming during the drug tolerant phase preceding acquired resistance through a genome-wide RNAi screen and comprehensive transcriptome analysis. This approach identified mRNA transport and translation pathways as key regulators of the metabolic response to BRAFi in BRAF V600E melanoma cells. The research showed that metabolic adaptation is linked to selective mRNA transport and translation of metabolic proteins essential for BRAFi sensitivity and resistance, including glucose transporters and oxidative phosphorylation (OXPHOS) enzymes. This translational reprogramming depends on the RNA processing kinase UHMK1, which regulates mitochondrial flexibility and the availability of metabolic proteins by controlling the export and translation of their mRNA. Notably, the inactivation of UHMK1 enhances sensitivity to combined BRAF and MEK inhibition, delaying resistance in multiple in vivo models. Overall, the findings suggest that selective mRNA transport and translation contribute to metabolic adaptation driving cancer cell plasticity during therapy and that targeting this pathway may delay resistance to MAPK-targeted therapies [145]. In 2021, it was proposed that cyclin-dependent kinase 4 and 6 (CDK4/6) inhibitors could be utilized with success for breast cancer, in response to metabolic reprogramming observed in melanoma following standard BRAF/MEK inhibition implicated in both therapeutic response and resistance. Preclinical studies have shown that CDK4/6 inhibitors can overcome resistance to BRAF/MEKi, resulting in sustained tumor regression. In this study, the authors observed that despite CDK4/6 inhibition not significantly affecting the metabolic changes from BRAF/MEK therapy, it independently enhances mitochondrial metabolism. This increase is partly driven by glutamine metabolism and fatty acid oxidation, with partial dependence on p53. These findings suggest novel p53-dependent metabolic vulnerabilities that could be targeted to enhance the effectiveness of CDK4/6 inhibitors [146]. Soumoy, L. and collaborators investigated both 1H-NMR-based metabonomic and protein microarrays, cell lines with mutant BRAF, NRAS, or cKIT and with acquired resistances to BRAF, MEK, or cKIT inhibitors, respectively. Through this study, it emerged that melanoma cells treated with BRAFi showed metabolic interruptions due to acute treatment exposure but partially recuperated their metabolism after long-term exposure. The microarrays highlighted a decreased level of protein, connected to the drug efficacy at the level downstream part of the MAPK signaling pathway [147].

## 7. Strategies to Overcome BRAF Inhibitor Resistance in Melanoma

The discovery of the BRAF oncogene, along with the introduction of BRAFi into clinical practice, has dramatically improved survival rates for patients with BRAF-mutant melanoma, which accounts for approximately 40–60% of cases. Targeting the constitutively active BRAF V600 mutant protein, in association with MEKi to prevent the paradoxical reactivation of the BRAF pathway, represents one of the most effective therapeutic strategies in the management of this potentially lethal skin cancer.

### 7.1. Mechanistic Insights into Resistance

#### 7.1.1. Approved Targeted Therapies

To date, FDA and EMA have approved three different BRAFi (vemurafenib, dabrafenib, encorafenib) in combination with three MEKi (cobimetinib, trametinib, binimetinib) for use in the metastatic setting. Among these, only the combination of dabrafenib/trametinib is approved as adjuvant therapy in fully resected stage III BRAF V600-mutant melanoma subjects. Although doublet therapy is associated with high response rate and significant tumor shrinkage, this achievement is usually short, lasting no more than 11–14.9 months. Indeed, the occurrence of drug resistance develops in most patients, representing a major clinical challenge.

#### 7.1.2. Mechanism of Resistance

Resistance can be divided into primary (or intrinsic) and secondary (adaptative and acquired). Primary resistance refers to the tumor’s inability to respond from the onset of therapy, and it occurs in approximately 15% of patients, who do not achieve any clinical benefit. Among the mechanisms underlying this phenomenon, several genetic alterations have been identified, including loss of PTEN or NF1 genes, CCND1 gene amplification, COT (MAP3K8) overexpression RAC1 mutation, and the disruption of the USP28-FBW7 complex. Moreover, epigenetic and transcriptomic mechanisms can also be involved. Secondary resistance concerns the loss of tumor responsiveness to treatment during drug administration; it takes place in more than 80% of patients and can be divided into adaptive or acquired. While the first is often reversible and not due to mutation occurrence (e.g., reactivation of ERK1/2 pathway, RTKs, or MITF upregulation), the latter typically involves the development of irreversible mutations and heterogeneous molecular changes in multiple intracellular signaling pathways (e.g., development of BRAF gene amplification and splicing variants, NRAS or MEK1/2 mutations, RAF paradox and dimerization, RTK hyperactivation, or immune suppression mediated by NLRP1 inflammasome activation).

#### 7.1.3. Clinical Strategies to Overcome BRAFi/MEKi Resistance

Several strategies have been proposed to overcome resistance to BRAFi/MEKi, including alternative treatment schedules, combinations with immune checkpoint inhibitors (ICIs) or, more promisingly, with other targeted therapies. Based on preclinical models suggesting that a discontinuous dosing strategy could prevent the onset of lethal drug-resistant disease [148], this approach has been further investigated in clinical settings by two different phase II randomized studies that did not demonstrate a superior activity for the intermittent use compared to continuous administration. On the contrary, in the case of the dabrafenib/trametinib combination, the intermittent schedule was associated with a detrimental effect [149,150].

The combination of BRAFi and MEKi with immune checkpoint inhibitors (ICIs) has been explored with the goal of integrating the distinct advantages of both strategies: the rapid and deep responses of targeted therapy with the long-term survival advantages of immunotherapy. Preclinical studies suggested that BRAFi could enhance T-cell-mediated anti-tumor responses through several mechanisms, providing a rationale for this combined approach [125,151,152,153]. However, safety concerns, particularly with combinations involving anti-CTLA-4 agents like ipilimumab, have limited its clinical use due to significant toxicity [154,155]. Three international clinical trials explored this approach in unresectable/metastatic BRAF-mutant melanoma patients: the phase III IMspire 150 (atezolizumab + vemurafenib/cobimetinib vs. vemurabenib/cobimetinib) [156], the phase I/II KEYNOTE-022 (pembrolizumab + dabrafenib/trametinib vs. dabrafenib/trametinib) [157], and the phase III COMBI-i (spartalizumab + dabrafenib/trametinib vs. dabrafenib/trametinib) [158]. Despite the encouraging biological rationale, these clinical trials have shown conflicting and mostly inconsistent efficacy outcomes, with a higher incidence of toxicity for the experimental approaches with respect to the doublets. To date, only the atezolizumab-based triplet has been approved by FDA, but not by EMA, mirroring the debated safety profile and benefits above all considering the update on overall survival data recently published [159].

#### 7.1.4. Combining BRAFi/MEKi with Other Targeted Agents

Significant efforts and attempts have also been made to explore the addition of other targeted agents to BRAFi and MEKi to overcome resistance mechanisms, not limited to the following studies. Research conducted by Acquaviva and collaborators demonstrated that heat shock protein 90 (Hsp90) inhibition by ganetespib in BRAFV600E-mutant melanoma cell lines leads to the loss of mutant BRAF expression and a reduction in both MAPK and PI3K/AKT signaling. This mechanism exhibited greater in vitro potency and superior antitumor efficacy compared to BRAFi/MEKi. Furthermore, the combined inhibition of Hsp90 and BRAF V600E showed a synergistic effect in vemurafenib-sensitive melanoma models, yielding promising results both in vitro and in vivo, suggesting a potential therapeutic approach to enhance the efficacy of targeted treatments. A key aspect of this study is that ganetespib effectively overcame both intrinsic and acquired resistance to vemurafenib, with the latter characterized by ERK signaling reactivation. Additionally, the continuous suppression of BRAF V600E with vemurafenib increased the sensitivity to MEKi even after resistance had developed. While ganetespib treatment reduced but did not completely abolish baseline ERK activity, profiling studies revealed that adding a MEKi completely blocked ERK reactivation in resistant cells, with ganetespib demonstrating superior combinatorial activity compared to vemurafenib. Moreover, the combination of ganetespib and the MEKi TAK-733 induced tumor regression in vemurafenib-resistant xenograft models. Overall, these findings suggest that ganetespib could serve as an effective therapeutic option, both as a monotherapy and in combination, for treating BRAF(V600E)-mutant melanoma, particularly as a strategy to overcome acquired resistance to selective BRAFi [160].

Belli and collaborators demonstrated that in melanoma cell lines resistant to both BRAF and MEKi, the SAN and A375 cell lines, which harbor the BRAF-V600 mutation and were initially sensitive to these drugs, resistance was primarily driven by the overexpression of specific TKRs, including EphA2 and DDRs. Notably, their study showed that this resistance could be effectively countered by the multikinase inhibitor ALW-II-41-27, which not only reduced TKR expression but also blocked the AKT/mTOR and MAPK signaling pathways. The efficacy of ALW-II-41-27 was further supported by its ability to decrease cell viability, invasion, and migration in drug-resistant melanoma cells. This evidence suggests that targeting resistance mechanisms mediated by TKRs through ALW-II-41-27 or other multikinase inhibitors could represent an effective therapeutic strategy to overcome resistance in metastatic BRAF-mutant melanoma [161].

In another study, various combination therapies, including the metabolic modulator dichloroacetate (DCA), were examined to enhance treatment efficacy in melanoma cell lines. Given that the MAPK and PI3K/AKT/mTOR pathways are often dysregulated and interconnected in melanoma, and considering the role of the Warburg effect in influencing therapy response, the authors evaluated the impact of these treatments on the proliferation and survival of melanoma cells with different genetic profiles, suggesting that the combination of MAPK pathway inhibitors with mTOR pathway inhibitors and/or DCA should be considered as therapeutic options to treat melanoma patients, as the combinations potentiated the effects of each drug individually [162].

#### 7.1.5. Personalized Combination Therapies

As previously mentioned, although the BRAF V600E mutation represents the primary oncogenic driver, resistant tumors often rely on alternative signaling pathways. Therefore, understanding and targeting patient-specific altered signaling networks is crucial for improving therapeutic efficacy. In this context, a recent study proposes a personalized approach to combination therapy, based on the analysis of network alterations specific to each melanoma tumor. Using an information-theoretic model, the authors identify high-resolution, patient-specific, altered signaling signatures. These signatures consist of multiple interconnected subnetworks, all of which must be simultaneously targeted to effectively inhibit aberrant signaling. Based on these findings, the authors design personalized drug combinations, often utilizing FDA-approved drugs, with the goal of optimizing therapeutic strategies. This approach has been validated both in vitro and in vivo, showing that individualized treatment regimens, rationally designed according to the patient’s altered signaling profile, are more effective than the conventional targeted therapy (BRAFi/MEKi). Furthermore, the study highlights the high selectivity of these personalized drug combinations as a regimen effective for one BRAF V600E-mutant tumor may be less effective for another and vice versa. Therefore, the authors propose a clinically relevant model that could guide the rational design of patient-specific drug combinations for melanoma treatment, offering the potential to improve therapeutic outcomes and overcome drug resistance in a personalized manner [163].

In the context of personalized medicine and more specifically tailored therapies, a recent study by Colakoglu et al. characterized the encorafenib-resistant A375-MM cells, which were generated through an in vitro protocol after three months of drug exposure. A comprehensive analysis using WST-1 assays, Annexin V staining, cell cycle analysis, morphological studies, live-cell imaging, Western blotting, RNA sequencing (RNA-Seq), transmission electron microscopy (TEM), oxidative stress measurements, and iron quantification assays was performed. The results showed that the resistant cells had increased cell viability, reduced apoptosis, G0/G1 phase cell cycle arrest, and the upregulation of autophagy markers Beclin and LC3, as well as an increase in AKT signaling activity. Moreover, RNA-Seq analysis revealed epigenetic alterations associated with resistance, particularly involving ferritin family genes and ion transport pathways. Additionally, the elevated levels of NCOA4, FTH1, and intracellular iron in A375-R cells suggested that iron metabolism-related processes, such as ferritinophagy, may be activated. This hypothesis was further supported by TEM imaging and oxidative stress evaluations, which confirmed iron accumulation and metabolic adaptations. These findings highlight that targeting iron storage, transport, and ferritinophagy could be a promising therapeutic strategy to counteract encorafenib resistance [164].

#### 7.1.6. Future Directions

Future research on targeted therapy to improve melanoma management should aim at expanding the range of available targeted treatments and optimizing combination strategies, with an emphasis on personalizing patient care as much as possible [165]. Future strategies should focus on developing next-generation BRAFi characterized by enhanced selectivity and reduced propensity for resistance. Additionally, overcoming resistance mechanisms through multiple strategies will be crucial to prolonging therapeutic response. Advances in molecular biology, RNA-based therapies, and gene editing could offer new solutions to overcome drug resistance thanks to the development of (*i*) small interfering RNAs and microRNAs targeting key oncogenic pathways implicated in resistance mechanisms, (*ii*) long non-coding RNAs (lncRNAs) influencing melanoma progression and therapy resistance, and (*iii*) CRISPR-based genome editing to correct resistance-associated mutations in genes such as BRAF, MEK, or NRAS [166,167,168].

### 7.2. Biomarker-Guided Strategies and Emerging Therapeutic Approaches

A clear mechanistic understanding of both adaptive and acquired resistance lays the foundation for developing dynamic, biomarker-guided interventions.

#### 7.2.1. Real-Time Biomarkers and Longitudinal Monitoring

The introduction of combined BRAF and MEK inhibition dramatically improves survival outcomes; however, despite initial tumor regression, most patients eventually relapse, and median progression-free survival rarely exceeds 12–18 months [169]. 

A key component in addressing this challenge lies in the development and validation of biomarkers that can guide therapeutic decisions in real time. Genomic alterations such as secondary NRAS mutations, NF1 loss, PTEN alterations, and BRAF splice variants are consistently linked to both adaptive and acquired resistance, providing actionable targets for rational combination approaches [170]. Beyond genomic data, transcriptomic and epigenetic profiling reveal how melanoma cells transition between drug-tolerant and fully resistant states. These signatures help to distinguish adaptive responses—which are often reversible and dependent on transcriptional reprogramming—from acquired resistance, which typically involves stable genetic alterations [171]. Circulating tumor DNA (ctDNA) emerges as a particularly powerful tool for longitudinal monitoring. Rising ctDNA levels or the early detection of resistance-associated mutations frequently precede radiologic progression, creating a window to adjust therapy before clinical deterioration occurs [172]. Similarly, immune-related biomarkers, including shifts in tumor-infiltrating lymphocytes (TILs) and PD-L1 expression during BRAFi/MEKi treatment, predict which patients are most likely to benefit from the early integration of immune checkpoint inhibitors [173].

#### 7.2.2. Targeting Bypass Signaling and Pathway Reactivation

To counter resistance, multi-pronged therapeutic strategies are being explored. Dual and triple pathway inhibition gain momentum as a way to prevent bypass signaling. Combinations of BRAFi/MEKi with PI3K/AKT/mTOR or ERK inhibitors show promise in preclinical and early-phase clinical studies aimed at blocking MAPK pathway reactivation Targeting upstream RAS signaling with agents such as SHP2 or SOS1 inhibitors offers another approach, particularly in NRAS-mutant or NF1-deficient tumors [174].

#### 7.2.3. Integrating Targeted Therapy and Immunotherapy

The integration of targeted therapy with immunotherapy is another avenue under active investigation. BRAFi/MEKi treatment induces a transient increase in tumor antigen presentation and T-cell infiltration, creating a window of opportunity for immune engagement [175]. Trials testing sequential or concurrent BRAFi/MEKi and PD-1/PD-L1 inhibitors seek to harness this synergy, while adaptive strategies that switch patients to immunotherapy upon ctDNA-defined molecular progression are being evaluated as precision-based interventions [176].

#### 7.2.4. Tumor Microenvironment and Metabolic Vulnerabilities

The tumor microenvironment (TME) plays a pivotal role in supporting resistant clones. Interactions with cancer-associated fibroblasts and immunosuppressive myeloid cells sustain MAPK signaling and foster drug tolerance. Novel agents targeting FAK signaling, stromal remodeling, and myeloid cell reprogramming are currently being tested to disrupt these TME-driven resistance mechanisms [177]. Metabolic targeting is also emerging as a promising field. BRAFi-resistant melanoma cells frequently rewire their metabolism, shifting toward mitochondrial oxidative phosphorylation (OXPHOS). This adaptation creates a therapeutic vulnerability, and inhibitors of OXPHOS or glutamine metabolism are now being explored in combination with BRAFi/MEKi to dismantle these resistant networks [178].

#### 7.2.5. From Fixed Sequences to Adaptive Algorithms

For clinical practice, these advances emphasize the transition from fixed treatment sequences to adaptive, biomarker-guided algorithms. Routine integration of ctDNA and multi-omic profiling into clinical workflows has the potential to enable early detection of resistance and inform timely therapeutic changes [179]. At the policy level, embedding longitudinal molecular monitoring into standard-of-care protocols and promoting adaptive trial designs are crucial to accelerate translation. Future research focuses on validating triple combination regimens, exploiting metabolic vulnerabilities, and integrating TME-modulating agents in biomarker-selected populations. The application of artificial intelligence and computational oncology to synthesize longitudinal data is poised to drive the next step toward truly personalized melanoma care [5].

## 8. Discussion, Conclusions, and Future Directions

The introduction of BRAFi, particularly when combined with MEKi, has transformed the treatment landscape for patients with BRAF V600-mutant melanoma. Nevertheless, despite the substantial initial clinical benefit, resistance—both intrinsic and acquired—continues to compromise long-term disease control. Our review underscores the multifactorial nature of resistance, involving not only genetic reactivation of the MAPK pathway but also parallel signaling activation (e.g., PI3K/AKT), immune evasion, and metabolic reprogramming. Dual inhibition of BRAF and MEK has provided improved survival outcomes compared to monotherapy, yet the near-universal emergence of secondary resistance reveals the limitations of current pathway-centric approaches. This phenomenon reflects the inherent plasticity of melanoma cells and the adaptability of the tumor microenvironment, necessitating a more nuanced therapeutic paradigm.

Several clinical implications emerge from these insights. First, resistance mechanisms often evolve dynamically and heterogeneously, underscoring the need for longitudinal molecular monitoring, ideally through minimally invasive tools such as circulating tumor DNA. Integrating such technologies into clinical practice could enable early detection of resistance trajectories and guide timely therapeutic interventions. Moreover, given the inter-patient variability in resistance patterns, treatment must increasingly be personalized, grounded in multi-omic and immune profiling. From a policy standpoint, these findings argue for systemic changes to support the routine implementation of precision oncology. This includes infrastructure for genomic profiling, reimbursement policies for advanced diagnostics, and digital platforms that facilitate multidisciplinary data interpretation. A commitment to equitable access to personalized therapies is essential to avoid widening disparities in melanoma outcomes.

On the research front, the review highlights several unmet needs. For example, sequencing and scheduling strategies for targeted and immune therapies remain poorly defined. While combinations of BRAF/MEKi with immune checkpoint blockade hold promise, they are complicated by overlapping toxicities, unclear patient selection criteria, and a lack of consensus on optimal timing. There is also a growing recognition that metabolic rewiring is not just a bystander phenomenon but a resistance driver and potentially a therapeutic target. Investigating agents that target mitochondrial function, oxidative stress balance, or nutrient scavenging mechanisms may open new frontiers in therapy. More broadly, the future of melanoma care will likely shift toward adaptive therapeutic strategies, guided by real-time data integration and modeling. This vision requires a confluence of translational science, computational oncology, and AI-based platforms capable of handling the complexity of tumor evolution. The design of clinical trials must also evolve to become more flexible and biomarker-driven, allowing for iterative hypothesis testing and rapid incorporation of emerging findings. In summary, overcoming BRAFi resistance will require moving beyond static interventions and toward a dynamic, systems-level approach to cancer treatment. This includes targeting not only tumor-intrinsic pathways but also the microenvironment and metabolic dependencies and doing so in a way that is responsive to the evolving molecular landscape of each patient’s disease. By embracing this complexity, future melanoma therapies can become more durable, personalized, and aligned with the real-world challenges of clinical oncology.

## Figures and Tables

**Figure 1 pharmaceuticals-18-01235-f001:**
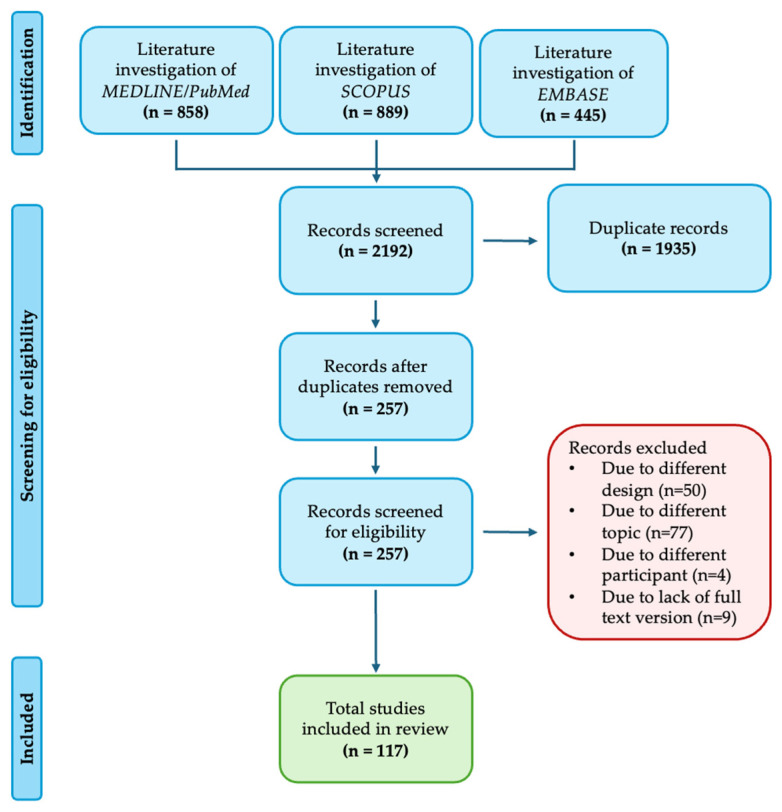
PRISMA flow diagram.

**Figure 2 pharmaceuticals-18-01235-f002:**
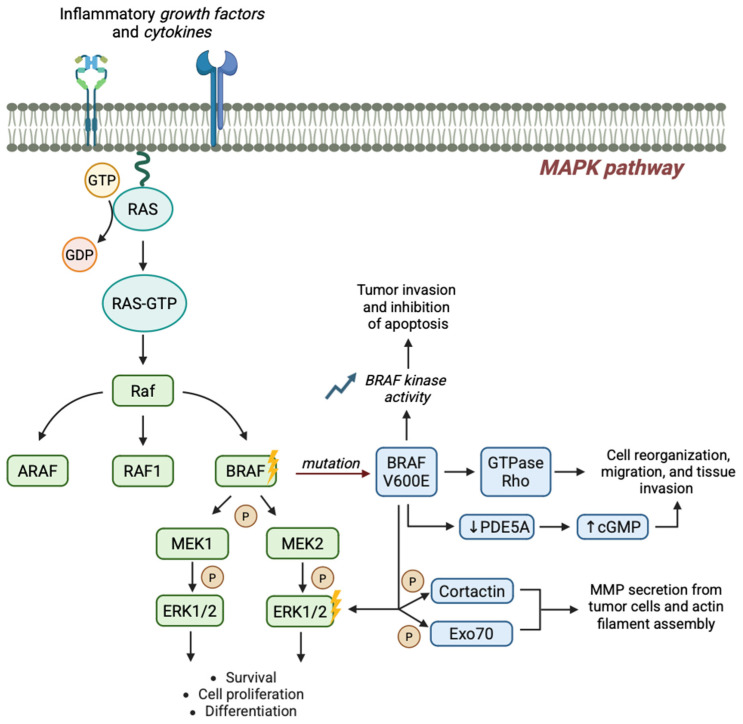
BRAF V600E-driven signaling promotes melanoma initiation and progression. The BRAF V600E mutation constitutively activates the MAPK pathway, promoting tumor growth and invasion by multiple signaling interference.

**Figure 3 pharmaceuticals-18-01235-f003:**
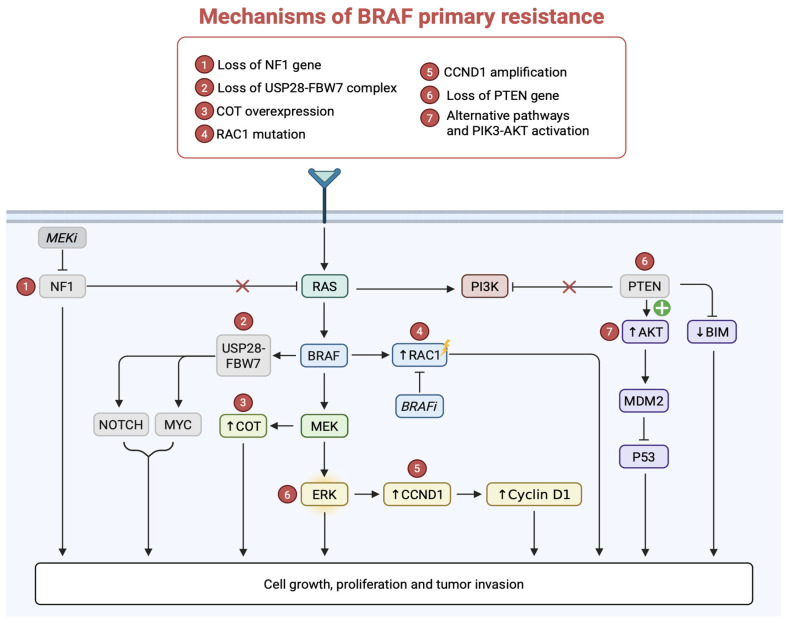
Molecular mechanisms underlying primary resistance in BRAF mutant melanoma. Mechanisms of primary resistance to BRAFi include MAPK reactivation, PI3K-AKT signaling, cyclin D1 upregulation, and loss of several genes (NF1, PTEN) promoting tumor cell proliferation and survival.

**Figure 4 pharmaceuticals-18-01235-f004:**
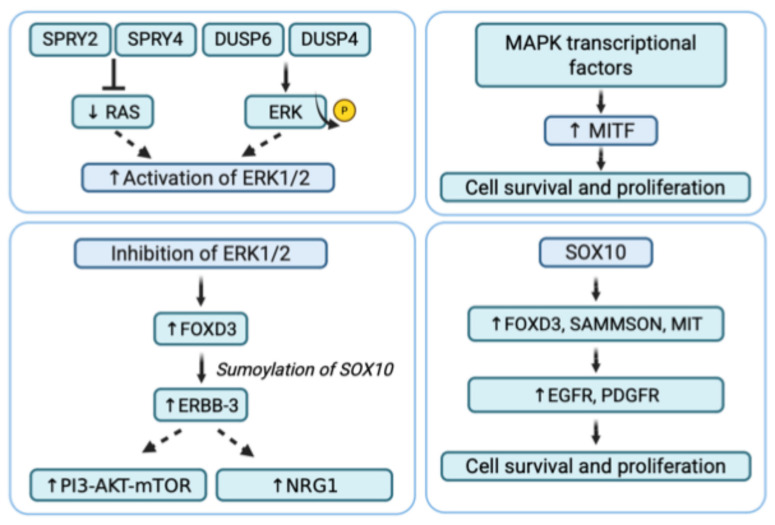
Secondary adaptive resistance in BRAF-mutant melanoma. Mechanisms of secondary resistance to BRAFi include MAPK reactivation, SOX10 upregulation, inhibition of ERK1/2, and upregulation of MITF, promoting cell survival and proliferation.

**Figure 5 pharmaceuticals-18-01235-f005:**
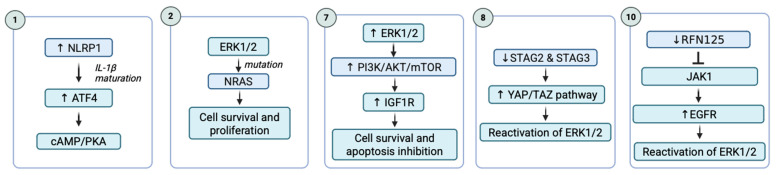
Secondary acquired resistance in BRAF-mutant melanoma. Mechanisms of secondary resistance to BRAFi include in this image: NLRP1 in MAPK/ERK pathway, NRAS mutations, PI3K/AKT pathway, STAG2 or STAG3 expression and YAP/TAZ pathway, and expression of ring finger protein 125 (RNF125).

**Table 1 pharmaceuticals-18-01235-t001:** FDA and EMA indications for targeted therapy in melanoma patients.

BRAFi + MEKi	Setting	Oral Administration	Registrative Studies	FDA Approval
Vemurafenib + Cobimetinib	Advanced melanoma	960 mg BID + 60 mg QD	coBRIM [11]	2011; 2015
Dabrafenib + Trametinib	Advanced melanoma	150 mg BID + 2 mg QD	COMBI-d [9] COMBI-v [10]	2013; 2018
Encorafenib + Binimetinib	Advanced melanoma	450 mg QD + 45 mg BID	COLUMBUS [8]	2018

**Table 2 pharmaceuticals-18-01235-t002:** Key landmark studies in primary resistance: genetic mechanisms of resistance.

References	Resistance Mechanism	Experimental Model	Key Findings
[28]	RAC1 mutation	Study on RAC1 mutations in melanoma	Identification of RAC1 as a therapeutic target in resistant melanomas
[29]	MMAC1/PTEN mutation	Identification of MMAC1, a tumor suppressor gene at 10q23.3	Mutations in MMAC1 in advanced cancers, including melanoma
[30]	MAPK/PI3K inhibition	Study of resistance pathways in melanoma treated with targeted therapies	Acquired resistance to therapy due to activation of YAP1 pathway
[31]	Loss of PTEN	Review of biochemical and clinical implications of PTEN loss	Resistance to melanoma treatments following PTEN loss
[32]	Loss of PTEN	Preclinical study in melanoma models	PTEN loss promotes the development of malignant melanoma
[33]	Loss of PTEN	Study on molecular mechanisms of BRAF inhibitor resistance	Resistance to BRAF inhibitors due to PTEN loss
[34]	CCND1 amplification	Profiling of CCND1 amplification in melanoma	Identification of a melanoma subtype with poor prognosis
[35]	Mutations in NF1 and RAS	Exome sequencing in sun-exposed melanomas	Identification of recurrent mutations in NF1 and RAS-related genes
[36]	Mutations in RAS	Analysis of RAS pathway regulation in melanoma	Description of RAS pathway regulation and its impact on resistance
[37]	Mutations in RAF pathway	Genetic sequencing in melanoma resistant to RAF inhibitors	Analysis of clinical resistance to RAF inhibitors
[38]	BRAF and NRAS mutation	Review on resistance mechanisms in BRAF- and NRAS-mutated melanomas	Multiple resistance pathways in BRAF/NRAS mutated melanomas
[39]	RAC1 mutation	Study on melanomas with RAC1 mutations	Characterization and outcomes of RAC1 mutated melanomas
[40]	Loss of USP28	Study on resistance mechanisms mediated by USP28 loss	USP28 loss drives resistance to RAF inhibitors

**Table 3 pharmaceuticals-18-01235-t003:** (a) Key landmark studies in primary resistance: epigenetic and transcriptomic mechanisms. (b) Key landmark studies in primary resistance: epigenetic and transcriptomic mechanisms. (c) Key landmark studies in primary resistance: epigenetic and transcriptomic mechanisms.

References	Resistance Mechanism	Experimental Model	Key Findings
(a)			
[41]	Epigenetic escape from BRAF dependency	Review of epigenetic mechanisms	Epigenetic alterations contribute to therapy resistance in BRAF-mutant melanoma
[42]	Persister cells	Experimental in vitro/in vivo	Persister cells show vulnerability to GPX4 inhibition
[43]	Phenotypic plasticity	Theoretical biology review	Phenotypic plasticity is an adaptive feature in tumor evolution
[44]	MITF-related resistance	Functional studies	MITF regulates genes controlling proliferation/invasiveness
[45]	Melanocyte biology	Review	Background on melanocyte function and pigmentation pathways
[46]	Endothelin receptor-mediated resistance	Functional, signaling pathway analysis	Blocking endothelin signaling reduces heterogeneity-driven resistance
[47]	Dedifferentiation-mediated resistance	Single-cell profiling	Reversible dedifferentiation leads to RAF inhibitor resistance
[48]	MITF/AXL ratio	Transcriptomic profiling	Low MITF/high AXL linked to resistance to targeted therapies
[49]	Cell state and signaling interaction	Review	MITF and IFNγ pathways modulate resistance-related cell states
[50]	Multiple resistance trajectories	Longitudinal single-cell studies	Multiple evolutionary paths lead to drug resistance
[51]	Rare cell variability	Single-cell transcriptomics	Rare subpopulations and reprogramming fuel resistance
[52]	Epigenetic modification enzymes	Biochemical assays	TET enzymes mediate DNA demethylation, impacting gene regulation
[53]	TET protein activity	Biochemistry/molecular biology	TET proteins modify methylcytosine, impacting epigenetic state
[54]	Aberrant methylation	Review	Methylation changes as biomarkers and therapeutic targets
[55]	PTEN methylation	Methylation analysis	PTEN methylation associated with prognosis
[56]	PTEN silencing	Promoter methylation assays	PTEN is epigenetically inactivated in melanoma
[57]	Global hypomethylation	Genome-wide methylation analysis	Melanomas show reduced global 5-mC levels
[58]	INK4a locus	Genetic knockout models	INK4a critical for cell cycle control and tumor suppression
[59]	P16 methylation	Methylation-specific PCR	P16 frequently methylated in NRAS-mutant melanomas
(b)			
[60]	EGFR pathway activation via epigenetic reprogramming	DNA methylation profiling, expression analysis, functional assays	A cryptic transcript of TBC1D16 activated by hypomethylation promotes melanoma progression through EGFR pathway activation.
[61]	Epigenetic escape from senescence via H3K9 demethylases	CRISPR-Cas9 screening, pharmacological inhibition, transcriptomics	Multiple H3K9 demethylases cooperate to suppress senescence and support melanoma cell proliferation; targeting them restores senescence.
[62]	Loss of senescence as an epigenetic switch	Review (summary of epigenetic changes from senescence to melanoma)	Highlights the reversibility and significance of senescence bypass in melanoma development and its epigenetic regulation.
[63]	Slow-cycling cell subpopulation promoting drug tolerance	In vivo melanoma xenografts, BrdU labeling, flow cytometry	Identifies a JARID1B+ slow-cycling subpopulation critical for sustained melanoma growth and therapy resistance.
[64]	Multidrug tolerance via stress-induced innate immunity	Transcriptomics, cell viability assays, in vivo models	Stress responses activate reversible drug-tolerant states; inhibition of inflammatory signaling pathways restores drug sensitivity.
[65]	Resistance to MAPKi via IGF pathway activation (SIRT6 loss)	SIRT6 knockdown, signaling pathway analysis, BRAF inhibitor response	SIRT6 deficiency activates IGF signaling, driving resistance to MAPK-targeted therapy in BRAF-mutant melanoma.
[66]	Not melanoma-focused; background reference	Microarray gene expression profiling	Establishes molecular subtypes of breast cancer, foundational for molecular classification methods applied in other cancers including melanoma.
[67]	Transcriptional dysregulation	Literature review	Proposes targeting basal transcription components (e.g., general transcription factors) as a cancer therapy approach.
[68]	MITF amplification conferring survival advantage	Genomic copy number analysis, gene expression, functional validation	MITF amplification promotes melanoma cell survival; MITF functions as a lineage-specific oncogene.
(c)			
[69]	Dysregulation of transcriptional programs	Literature review, functional genomics	Highlights key transcription factors and epigenetic modulators contributing to melanoma resistance and progression
[70]	MITF overexpression and plasticity	Molecular biology, transcriptional analysis	MITF functions as a lineage-specific oncogene involved in survival and resistance pathways
[71]	Germline MITF mutation (E318K)	Genetic sequencing, family-based studies	Mutation increases melanoma susceptibility and may alter transcriptional responses to therapy
[72]	MITF activity modulation via SUMO	Promoter-reporter assays, mutagenesis	SUMO modification influences promoter-specific MITF function, potentially affecting resistance profiles
[73]	MAPK signaling influences MITF	Signal transduction studies	P38 MAPK regulates MITF in response to external signals, linking environmental cues to transcriptional resistance
[74]	SOX10 as a lineage-specific TF	Gene sequencing, transactivation assays	SOX10 plays a key role in melanocyte lineage and can contribute to resistance via transcriptional control
[75]	Post-translational regulation of SOX10	Phospho-proteomics, cell-based assays	ERK regulates SOX10 via phosphorylation and SUMOylation, linking BRAF signaling to transcriptional responses
[76]	Mutant p53 gain-of-function	Review of p53 biology	Mutant p53 can acquire oncogenic functions, contributing to therapy resistance in diverse cancers including melanoma
[77]	Metabolic stress-mediated resistance	Molecular cell biology	PGC-1α supports survival under metabolic stress via modulation of p53, promoting resistance pathways
[78]	Aberrant p53 modifications	Review	Highlights how PTMs of p53 affect its tumor-suppressive functions, leading to resistance
[79]	Cross-talk among signaling pathways	Bioinformatics and gene expression profiling	Resistance to BRAF/MEK inhibitors linked to activity in additional cancer-related pathways
[80]	Apoptosis evasion	Drug screening, in vitro studies	Co-targeting Bcl2 and Mcl1 can overcome resistance and induce apoptosis in melanoma cells
[81]	Anti-apoptotic bias toward MCL1	Functional genomics, drug assays	MCL1 dependence confers resistance; its inhibition sensitizes cells to MAPK inhibitors
[82]	Apoptosis inhibition	CRISPR knockout screens	BCL-XL and MCL1 are essential for melanoma survival, representing resistance mechanisms
[83]	miR-211 loss and BRAF inhibitor sensitivity	MicroRNA profiling, metabolic assays	Loss of miR-211 increases vulnerability to metabolic stress and enhances BRAF inhibitor response
[84]	miR-211 reprograms ERK signaling	In vivo mouse models, miRNA overexpression	miR-211 promotes resistance by reactivating ERK signaling via DUSP6-ERK5 axis
[85]	Dual role of miR-211 in resistance and sensitivity	Literature synthesis	miR-211 can either promote or inhibit resistance depending on cellular context
[86]	miR-10b linked to aggressive melanoma	qRT-PCR in patient serum	miR-10b is upregulated in melanoma and associated with poor outcomes and possible resistance
[87]	miR-10b as a biomarker of resistance and progression	Clinical biomarker study	miR-10b expression correlates with melanoma aggressiveness and prognosis

**Table 4 pharmaceuticals-18-01235-t004:** Key landmark studies in secondary resistance.

References	Resistance Mechanism	Experimental Model	Key Findings
[76]	Dysregulation of transcriptional programs	Literature review, functional genomics	Highlights key transcription factors and epigenetic modulators contributing to melanoma resistance and progression
[77]	MITF overexpression and plasticity	Molecular biology, transcriptional analysis	MITF functions as a lineage-specific oncogene involved in survival and resistance pathways
[78]	Germline MITF mutation (E318K)	Genetic sequencing, family-based studies	Mutation increases melanoma susceptibility and may alter transcriptional responses to therapy
[79]	MITF activity modulation via SUMO	Promoter-reporter assays, mutagenesis	SUMO modification influences promoter-specific MITF function, potentially affecting resistance profiles
[80]	MAPK signaling influences MITF	Signal transduction studies	P38 MAPK regulates MITF in response to external signals, linking environmental cues to transcriptional resistance
[81]	SOX10 as a lineage-specific TF	Gene sequencing, transactivation assays	SOX10 plays a key role in melanocyte lineage and can contribute to resistance via transcriptional control
[82]	Post-translational regulation of SOX10	Phospho-proteomics, cell-based assays	ERK regulates SOX10 via phosphorylation and SUMOylation, linking BRAF signaling to transcriptional responses
[83]	Mutant p53 gain-of-function	Review of p53 biology	Mutant p53 can acquire oncogenic functions, contributing to therapy resistance in diverse cancers including melanoma
[84]	Metabolic stress-mediated resistance	Molecular cell biology	PGC-1α supports survival under metabolic stress via modulation of p53, promoting resistance pathways
[85]	Aberrant p53 modifications	Review	Highlights how PTMs of p53 affect its tumor-suppressive functions, leading to resistance
[86]	Cross-talk among signaling pathways	Bioinformatics and gene expression profiling	Resistance to BRAF/MEK inhibitors linked to activity in additional cancer-related pathways
[87]	Apoptosis evasion	Drug screening, in vitro studies	Co-targeting Bcl2 and Mcl1 can overcome resistance and induce apoptosis in melanoma cells
[88]	Anti-apoptotic bias toward MCL1	Functional genomics, drug assays	MCL1 dependence confers resistance; its inhibition sensitizes cells to MAPK inhibitors
[89]	Apoptosis inhibition	CRISPR knockout screens	BCL-XL and MCL1 are essential for melanoma survival, representing resistance mechanisms
[90]	miR-211 loss and BRAF inhibitor sensitivity	MicroRNA profiling, metabolic assays	Loss of miR-211 increases vulnerability to metabolic stress and enhances BRAF inhibitor response
[91]	miR-211 reprograms ERK signaling	In vivo mouse models, miRNA overexpression	miR-211 promotes resistance by reactivating ERK signaling via DUSP6-ERK5 axis
[92]	Dual role of miR-211 in resistance and sensitivity	Literature synthesis	miR-211 can either promote or inhibit resistance depending on cellular context
[93]	miR-10b linked to aggressive melanoma	qRT-PCR in patient serum	miR-10b is upregulated in melanoma and associated with poor outcomes and possible resistance
[94]	miR-10b as a biomarker of resistance and progression	Clinical biomarker study	miR-10b expression correlates with melanoma aggressiveness and prognosis

**Table 5 pharmaceuticals-18-01235-t005:** Key landmark studies in immune mechanisms of resistance.

References	Resistance Mechanism	Experimental Model	Key Findings
[124]	Loss of RNF125 leads to JAK1 deregulation	Cell models, Western blot, gene silencing	Loss of RNF125 promotes BRAF inhibitor resistance through increased JAK1 signaling
[125]	TNF from macrophages induces resistance	In vitro models, cytokine profiling	Macrophage-derived TNF sustains survival signals during MAPK inhibitor treatment
[126]	Microenvironment modulation	Patient tumor biopsies, gene expression	BRAF inhibition promotes a more immune-permissive tumor environment
[127]	Immune modulation	Literature review	BRAF inhibitors affect tumor immunity; rationale for combining with immunotherapy
[128]	MAPK inhibition impacts cytokine regulation	In vitro cytokine assays	JNK and p38 inhibition reduces IL-10, affecting immune responses
[129]	Immune cell recruitment post-inhibition	Mouse models, immunotherapy studies	BRAF inhibition increases CD8+ T-cell infiltration and improves immunotherapy
[130]	MAPK signaling suppresses immune recognition	siRNA, antigen presentation assays	BRAF-MAPK pathway contributes to immune evasion via MHC downregulation
[131]	Antigen presentation suppression	Imaging, immunology assays	BRAFV600E promotes MHC-I internalization to evade CD8+ T-cell killing
[132]	Immune modulation by mutation/inhibition	Patient samples, immune profiling	BRAF mutations/inhibition alter immune activity and antigen expression
[133]	T-cell recruitment via MAPK inhibition	Patient tumor samples, IHC	BRAF inhibitors lead to increased T-cell infiltration in metastatic melanoma
[134]	Synergy with immune checkpoint therapy	Preclinical models, combination treatments	Combining BRAF inhibition with PD-1/CTLA-4 blockade improves tumor control
[135]	DC function recovery with dual inhibition	Dendritic cell assays, melanoma models	Dual BRAF/MEK inhibition reverses DC dysfunction in BRAFV600E melanoma
[136]	NK cell involvement in treatment efficacy	NK cell depletion studies	NK cells necessary for full therapeutic response to BRAF inhibitors
[137]	IL-1-mediated immunosuppression	Cytokine assays, co-culture models	BRAF mutation increases IL-1, promoting immunosuppressive stroma
[138]	CCL2-mediated immune modulation	In vivo models, cytokine analysis	CCL2 promotes both tumor growth and immune surveillance, showing dual roles
[139]	TAMs as M2-polarized immunosuppressive cells	Literature review	M2 macrophages support tumor progression and inhibit immune responses
[140]	Macrophage polarization spectrum	Review	Discusses plasticity of macrophage activation from pro- to anti-inflammatory roles

**Table 6 pharmaceuticals-18-01235-t006:** Key landmark studies in metabolic pathway of resistance.

References	Resistance Mechanism	Experimental Model	Key Findings
[141]	MAPK-driven PD-L1 upregulation	In vitro assays, pharmacologic inhibition	MAPK reactivation promotes immune evasion via PD-L1 expression, reversible by MEK/PI3K inhibition
[142]	Warburg effect/metabolic adaptation	Review	Explains how altered cancer metabolism supports proliferation, indirectly linked to resistance mechanisms
[143]	Multiple, including microenvironmental influences	Review	Emphasizes holistic view of metastasis and resistance, including immune and stromal interactions
[144]	Multiple pathways (MAPK reactivation, metabolic rewiring)	Literature review	Comprehensive overview of mechanisms of resistance to BRAF inhibitors
[145]	PGC1α-mediated mitochondrial adaptation	Gene expression, oxidative stress assays	High PGC1α defines tumors resistant to oxidative stress, associated with poor therapy response
[146]	Metabolic adaptation through translational reprogramming	Transcriptomics, metabolic assays	Melanoma cells reprogram metabolism at the translational level to evade BRAF-targeted therapy
[147]	p53-dependent mitochondrial reprogramming	Pharmacologic inhibition, metabolic profiling	CDK4/6 inhibitors rewire metabolism via p53, suggesting metabolic vulnerabilities in resistant cells

## Data Availability

Not applicable.

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
