# Peer review of "Molecular Basis of BRAF Inhibitor Resistance in Melanoma: A Systematic Review"

_pharmaceuticals, 2025, doi:10.3390/ph18081235_

Round 1

Reviewer 1 Report

Comments and Suggestions for Authors

Please find below my peer review of the manuscript titled “Molecular Basis of BRAF Inhibitor Resistance in Melanoma: A Systematic Review” (Manuscript ID: Pharmaceuticals-3704525), submitted to Pharmaceuticals. I have carefully evaluated the manuscript in terms of scientific content, clarity, structure, and relevance.

The manuscript presents a thorough and well-structured overview of the molecular mechanisms underlying resistance to BRAF inhibitors in melanoma. It is scientifically sound, methodologically robust, and of high relevance to the oncology and pharmacology community. The categorization of resistance mechanisms and integration of recent literature are commendable.

However, some editorial and structural revisions are needed to enhance clarity and reader engagement. I have classified my suggestions into major and minor comments as detailed below.

Major Comments:

  1. Improve formatting and presentation of tables and figures, especially Table 1–5 and Figure 2–3.
  • Table 1–5 (landmark studies)
  • Figure 2 & Figure 3
  • PRISMA flow diagram
  1. Avoid redundancy between the introduction and the BRAF signaling overview. The introduction and section 4 (“BRAF Overview”) contain overlapping explanations of MAPK signaling. Streamlining this content would reduce redundancy and improve flow.
  2. Expand discussion on clinical implications and biomarker-driven therapy. While resistance mechanisms are well-explained, more discussion is needed on how this knowledge translates to clinical practice, particularly:
  • Biomarkers guiding therapy selection
  • Specific therapeutic strategies in resistant cases
  1. Include a brief section on future directions and emerging therapeutic strategies. (e.g., targeting tumor heterogeneity, immune-TME interplay, epigenetic therapies) would increase the impact of the review.
  2. Standardize reference formatting in accordance with journal guidelines. The references in the text are bracketed with numbers but lack consistent formatting. Ensure the reference style matches the journal guidelines (likely Vancouver or similar).

Minor Comments:

  1. Simplify complex sentence structures for better readability. Example: "Adaptive resistance is characterized by a first response to BRAF inhibitor treatment that, , is lacking during..." — remove duplicate commas and clarify the sentence.
  2. Ensure consistency in the use of abbreviations and terminology (e.g., “BRAFi” vs “BRAF inhibitors”). Define abbreviations upon first use (e.g., “USP28-FBW7,” “NGFR”).
  3. Correct minor typographical and grammatical errors throughout (e.g., unnecessary line breaks, spacing).

Based on the above, I recommend minor revision of the manuscript.

Comments on the Quality of English Language
  1. Ensure consistency in the use of abbreviations and terminology (e.g., “BRAFi” vs “BRAF inhibitors”). Define abbreviations upon first use (e.g., “USP28-FBW7,” “NGFR”).
  2. Correct minor typographical and grammatical errors throughout (e.g., unnecessary line breaks, spacing).

Author Response

Comment 1: Please find below my peer review of the manuscript titled “Molecular Basis of BRAF Inhibitor Resistance in Melanoma: A Systematic Review” (Manuscript ID: Pharmaceuticals-3704525), submitted to Pharmaceuticals. I have carefully evaluated the manuscript in terms of scientific content, clarity, structure, and relevance. The manuscript presents a thorough and well-structured overview of the molecular mechanisms underlying resistance to BRAF inhibitors in melanoma. It is scientifically sound, methodologically robust, and of high relevance to the oncology and pharmacology community. The categorization of resistance mechanisms and integration of recent literature are commendable.

However, some editorial and structural revisions are needed to enhance clarity and reader engagement. I have classified my suggestions into major and minor comments as detailed below.

Answer to Comment 1: We thank the Reviewer for having appreciated our work

Comment 2: Improve formatting and presentation of tables and figures, especially Table 1–5 and Figure 2–3.; PRISMA flow diagram

Answer to Comment 2: We thank the Reviewer for this comment. We have improved the formatting and presentation of tables, Figures and PRISMA flow diagram

Comment 3: Avoid redundancy between the introduction and the BRAF signaling overview. The introduction and section 4 (“BRAF Overview”) contain overlapping explanations of MAPK signaling. Streamlining this content would reduce redundancy and improve flow.

Answer to Comment 3: We thank the Reviewer for this insightful comment. To address the redundancy between the Introduction and Section 4 (“BRAF Overview”), we have revised both sections to streamline the discussion of MAPK signaling. This restructuring reduces repetition and improves the overall flow of the manuscript

Comment 4: Expand discussion on clinical implications and biomarker-driven therapy. While resistance mechanisms are well-explained, more discussion is needed on how this knowledge translates to clinical practice, particularly: i) Biomarkers guiding therapy selection, ii) Specific therapeutic strategies in resistant cases

Answer to Comment 4: We thank the Reviewer for this valuable suggestion. In response, we have completely revised the previous chapter entitled “How shall we overcome resistance” to form the new chapter 7, entitled “Strategies to Overcome BRAF Inhibitor Resistance in Melanoma,” consisting of two sections: 7.1 Mechanistic Insights into Resistance and 7.2 Biomarker-Guided Strategies and Emerging Therapeutic Approaches, including all the subsections in which I have addressed your requests relating to comments 4 and 5, accompanying the whole with 11 new citations that have been added to the previous ones.

Comment 5: Include a brief section on future directions and emerging therapeutic strategies. (e.g., targeting tumor heterogeneity, immune-TME interplay, epigenetic therapies) would increase the impact of the review.

Answer to Comment 5: See the reply to the previous comment

Comment 6: Standardize reference formatting in accordance with journal guidelines. The references in the text are bracketed with numbers but lack consistent formatting. Ensure the reference style matches the journal guidelines (likely Vancouver or similar).

Answer to Comment 6: We thank the Reviewer for pointing this out. All references now follow the Vancouver-style, as specified by the journal.

Minor Comments:

  1. Simplify complex sentence structures for better readability. Example: "Adaptive resistance is characterized by a first response to BRAF inhibitor treatment that, , is lacking during..." — remove duplicate commas and clarify the sentence.
  2. Ensure consistency in the use of abbreviations and terminology (e.g., “BRAFi” vs “BRAF inhibitors”). Define abbreviations upon first use (e.g., “USP28-FBW7,” “NGFR”).
  3. Correct minor typographical and grammatical errors throughout (e.g., unnecessary line breaks, spacing).

Response to Minor Comments:

  1. We have revised the identified sentence and similar instances to simplify complex structures and improve readability.
  2. We have reviewed the manuscript for consistency in the use of abbreviations and terminology. All abbreviations (e.g., “USP28-FBW7,” “NGFR,” and “BRAFi”) are now defined upon first use, and terminology is standardized throughout the text.
  3. We have carefully proofread the manuscript and corrected typographical and grammatical errors, including unnecessary line breaks, inconsistent spacing, and formatting issues.

Comments on the Quality of English Language

  1. Ensure consistency in the use of abbreviations and terminology (e.g., “BRAFi” vs “BRAF inhibitors”). Define abbreviations upon first use (e.g., “USP28-FBW7,” “NGFR”).
  2. Correct minor typographical and grammatical errors throughout (e.g., unnecessary line breaks, spacing).

Response to Comments on the Quality of English Language

  1. We thank the Reviewer for this observation. We have revised the manuscript to ensure consistent use of abbreviations and terminology. All abbreviations, including “BRAFi,” “USP28-FBW7,” and “NGFR,” are now defined upon first mention and used consistently throughout the text

  1. We have proofread the manuscript to correct typographical and grammatical errors, including unnecessary line breaks, spacing inconsistencies, and other minor formatting issues, to improve overall clarity and readability.

Reviewer 2 Report

Comments and Suggestions for Authors

The manuscript provides a comprehensive overview of the molecular mechanisms driving resistance to BRAF inhibitors in BRAF-mutant melanoma, with additional reference to resistance mechanisms involving MEK inhibitors. The authors classify resistance into primary (intrinsic), adaptive, and acquired forms, and explore genetic, epigenetic, metabolic, and immune-related pathways implicated in therapeutic failure. The review synthesizes findings from 117 studies and presents them in a structured format, supported by tabular summaries and detailed mechanistic explanations. The authors also highlight emerging strategies to overcome resistance, including the use of combinatorial therapies and biomarker-guided approaches. Overall, the article is interesting and timely, addressing a clinically relevant challenge in melanoma therapy. The review is clearly written and well-structured. The categorization of resistance mechanisms is logical and facilitates understanding of a complex topic. I have the following questions and comments for the authors:

  • While Tables 1–5 provide valuable summaries of key studies related to resistance mechanisms in BRAF-mutant melanoma, their current placement within the manuscript may impede readability and comprehension. I recommend relocating each table to directly follow the corresponding sections of the mechanistic discussion in Section 6. Additionally, the manuscript would benefit from brief narrative references to each table in the text.

  • The information contained in Tables 1–5 is relevant and well-aligned with the review’s scope. However, I recommend removing full author names and article titles and replacing them with reference numbers or abbreviated citations (e.g., [45]). Additionally, standardizing the table structure and summarizing key findings and implications in a consistent column format would significantly improve clarity. I would recommend including the following columns: mechanisms of resistance, experimental model (e.g., cell line, clinical samples), key findings, and references.

  • I strongly encourage the authors to include a schematic illustration summarizing the molecular mechanisms of secondary resistance to BRAF and MEK inhibitors. Given the complexity and variety of the mechanisms described — including MAPK reactivation, NRAS mutations, RTK signaling, and transcriptomic shifts — a visual summary would significantly enhance clarity and understanding.

  • The manuscript currently discusses BRAFi and MEKi treatment in both the Introduction and Section 5, resulting in some redundancy. I suggest restructuring the Introduction to focus on the historical context, while transferring detailed therapeutic discussions — including drug mechanisms, combinations, and clinical outcomes — to Section 5. This restructuring would improve clarity and reduce repetition.

Author Response

Reviewer #2

Comment 1: The manuscript provides a comprehensive overview of the molecular mechanisms driving resistance to BRAF inhibitors in BRAF-mutant melanoma, with additional reference to resistance mechanisms involving MEK inhibitors. The authors classify resistance into primary (intrinsic), adaptive, and acquired forms, and explore genetic, epigenetic, metabolic, and immune-related pathways implicated in therapeutic failure. The review synthesizes findings from 117 studies and presents them in a structured format, supported by tabular summaries and detailed mechanistic explanations. The authors also highlight emerging strategies to overcome resistance, including the use of combinatorial therapies and biomarker-guided approaches. Overall, the article is interesting and timely, addressing a clinically relevant challenge in melanoma therapy. The review is clearly written and well-structured. The categorization of resistance mechanisms is logical and facilitates understanding of a complex topic.

Answer to Comment 1: We thank the Reviewer for the positive feedback. We are pleased to hear that the manuscript was found to be comprehensive, clearly written, and well-structured.

Comment 2: I have the following questions and comments for the authors. While Tables 1–5 provide valuable summaries of key studies related to resistance mechanisms in BRAF-mutant melanoma, their current placement within the manuscript may impede readability and comprehension. I recommend relocating each table to directly follow the corresponding sections of the mechanistic discussion in Section 6. Additionally, the manuscript would benefit from brief narrative references to each table in the text.

Answer to Comment 2: We thank the Reviewer for this suggestion. In response, we have relocated Tables 1–5 to follow the corresponding sections within the mechanistic discussion in Section 6 to improve readability and coherence. Additionally, we have added brief narrative references to each table in the main text to guide the reader and better integrate the tables into the flow of the discussion. We believe these changes enhance the clarity and accessibility of the information presented.

Comment 3: The information contained in Tables 1–5 is relevant and well-aligned with the review’s scope. However, I recommend removing full author names and article titles and replacing them with reference numbers or abbreviated citations (e.g., [45]). Additionally, standardizing the table structure and summarizing key findings and implications in a consistent column format would significantly improve clarity. I would recommend including the following columns: mechanisms of resistance, experimental model (e.g., cell line, clinical samples), key findings, and references.

Answer to Comment 3: In accordance with the suggestion, we have revised Tables 1–5 by removing full author names and article titles. Additionally, we have standardized the structure across all tables to include the recommended columns: mechanisms of resistance, experimental model, key findings, and references.

Comment 4: I strongly encourage the authors to include a schematic illustration summarizing the molecular mechanisms of secondary resistance to BRAF and MEK inhibitors. Given the complexity and variety of the mechanisms described — including MAPK reactivation, NRAS mutations, RTK signaling, and transcriptomic shifts — a visual summary would significantly enhance clarity and understanding.

Answer to Comment 4: To enhance clarity we have included a new schematic illustration summarizing the key molecular mechanisms of secondary resistance to BRAF and MEK inhibitors.

Comment 5: The manuscript currently discusses BRAFi and MEKi treatment in both the Introduction and Section 5, resulting in some redundancy. I suggest restructuring the Introduction to focus on the historical context, while transferring detailed therapeutic discussions — including drug mechanisms, combinations, and clinical outcomes — to Section 5. This restructuring would improve clarity and reduce repetition.

Answer to Comment 5: To reduce redundancy and improve clarity, we have restructured the manuscript by focusing the Introduction primarily on the historical context of BRAFi and MEKi treatments. Detailed discussions regarding drug mechanisms, combination therapies, and clinical outcomes have been consolidated and expanded in Section 5.

Round 2

Reviewer 2 Report

Comments and Suggestions for Authors

The authors have thoroughly addressed my comments and revised their manuscript accordingly.